# FGF18 alleviates hepatic ischemia-reperfusion injury via the USP16-mediated KEAP1/Nrf2 signaling pathway in male mice

Gaozan Tong[1,2,7], Yiming Chen[3,7], Xixi Chen[4,7], Junfu Fan[1], Kunxuan Zhu[1], ZiJing Hu[1], Santie Li[1], Junjie Zhu[1], Jianjun Feng[1], Zhaohang Wu[1], Zhenyu Hu[1], Bin Zhou[2], Litai Jin ⓘ[1], Hui Chen[1], Jingling Shen ⓘ[5] ✉, Weitao Cong ⓘ[1,2,6] ✉ & XiaoKun Li ⓘ[1,6] ✉

Hepatic ischemia-reperfusion injury (IRI) is a common complication occurs during hepatic resection and transplantation. However, the mechanisms underlying hepatic IRI have not been fully elucidated. Here, we aim to explore the role of fibroblast growth factor 18 (FGF18) in hepatic IRI. In this work, we find that Hepatic stellate cells (HSCs) secrete FGF18 and alleviates hepatocytes injury. HSCs-specific FGF18 deletion largely aggravates hepatic IRI. Mechanistically, FGF18 treatment reduces the levels of ubiquitin carboxyl-terminal hydrolase 16 (USP16), leading to increased ubiquitination levels of Kelch Like ECH Associated Protein 1 (KEAP1) and the activation of nuclear factor erythroid 2-related factor 2 (Nrf2). Furthermore, USP16 interacts and deubiquitinates KEAP1. More importantly, Nrf2 directly binds to the promoter of USP16 and forms a negative feedback loop with USP16. Collectively, our results show FGF18 alleviates hepatic IRI by USP16/KEAP1/Nrf2 signaling pathway in male mice, suggesting that FGF18 represents a promising therapeutic approach for hepatic IRI.

Hepatic ischemia-reperfusion injury (IRI) is an important cause of liver damage that occurs during clinical surgery such as liver resection, liver transplantation, and hemorrhagic shock, leading to graft failure, tissue injury, or even liver failure[1,2]. Oxidative stress is a key factor in the mechanisms underlying liver IRI. In the pathological process of liver IRI, injured hepatocytes overproduce reactive oxygen species (ROS) and attract Kupffer cells and other cells[3]. In response to oxidative stress, inflammatory cells immediately induce an inflammatory storm and eventually contribute to apoptosis of hepatocytes[4]. Thus, it is necessary to explore the mechanism of how to alleviate the level of ROS during IRI, in order to provide the new therapeutic insights into these clinical problems.

Fibroblast growth factor 18 (FGF18), a member of the exocrine fibroblast growth factor (FGF) family, plays a crucial role in organ development and pathological conditions[5], and widely participates in cell migration, proliferation, inflammation, and apoptosis[6,7]. For instance, it plays vital roles in lung alveolar development during late embryonic lung development stages, and also enhances the elastogenesis in lung myofibroblasts[8]. In addition, FGF18 is reportedly to be related to regulation of cartilage development and osteoarthritis.

[1]School of Pharmaceutical Science, Wenzhou Medical University, Wenzhou 325000, China. [2]Oujiang Laboratory (Zhejiang Lab for Regenerative Medicine, Vision and Brain Health), School of Pharmaceutical Science, Wenzhou Medical University, Wenzhou 325000, China. [3]Department of Hepatobiliary Surgery, The Second Affiliated Hospital and Yuying Children's Hospital of Wenzhou Medical University, Wenzhou, Zhejiang, China. [4]Department of pharmacy, Taizhou Central Hospital, Taizhou, Zhejiang, China. [5]Institute of Life Sciences, College of Life and Environmental Sciences, Wenzhou University, Wenzhou, Zhejiang, China. [6]Haihe Laboratory of Cell Ecosystem, School of Pharmaceutical Science, Wenzhou Medical University, Wenzhou, China. [7]These authors contributed equally: Gaozan Tong, Yiming Chen, Xixi Chen. ✉e-mail: jingling_shen@wzu.edu.cn; cwt97126@126.com; profxiaokunli@163.com

Recently, a phase II clinical assay indicated that recombinant human FGF18 protein effectively enhances the proliferation of chondrocytes and the production of extracellular matrix, suggesting it has good potential to treat joint diseases[9]. Furthermore, FGF18 participates in clear cell renal cell carcinoma proliferation and invasion by regulating the epithelial-mesenchymal transition[10]. However, the role of FGF18 in the liver, particularly in response to hepatic IRI, has not been fully elucidated.

Ubiquitin carboxyl-terminal hydrolase 16 (USP16, also called UBPM), a widely expressed deubiquitinase, has been implicated to function in cell proliferation, division, and differentiation[11, 12]. It specifically deubiquitinates H2A histones, and thereby regulates gene transcription and DNA repair[13]. Usp16 knockout (KO) is embryonically lethal in mice, but does not affect the viability or identity of embryonic stem cell, with different levels of differentiation[14]. Moreover, differentiation of embryonic stem cells largely depends on USP16-mediated deubiquitination of H2A histones[15]. USP16 also mediates disease progression without affecting ubiquitinated H2A histone levels. USP16 functions as a deubiquitinase of IKKβ with K33-linked ubiquitination and promotes its interaction with p105, contributing to the activation of the inflammatory response[16]. However, the function of USP16 in hepatic IRI remains unknown.

Nuclear factor erythroid 2-related factor 2 (Nrf2) is an important transcription factor that regulates the expression of antioxidant defense and detoxification genes in the liver and plays a key role not only in redox homeostasis, but also in drug metabolism, DNA repair, and regulation of mitochondrial function[17–19]. Under normal conditions, Nrf2 is bound to KEAP1 and retained in the cytoplasm, and KEAP1 inhibits the transcriptional activity of Nrf2 through ubiquitination and proteasomal degradation. Upon cell damage, Nrf2 dissociates from KEAP1 and enters the nucleus, where it binds to ARE elements and promotes the downstream antioxidant genes, such as those encoding heme oxygenase (HO)−1, NQO-1, and superoxide dismutase 2 (SOD2)[20]. Nrf2-mediated innate proinflammatory responses in macrophages were reported to attenuate liver IRI by binding to the promoter of Timp3 and inhibit the RhoA/ROCK pathway[21]. Billiar et al. found that IRG1, a metabolite of the tricarboxylic acid cycle, inhibits oxidative stress by activating the KEAP1-Nrf2-ARE signaling pathway and ultimately alleviates hepatic IRI[22]. These above studies suggest that Nrf2 signaling pathway can alleviate oxidative stress during hepatic IRI, and that targeting of this pathway may be therapeutic strategy for hepatic IRI.

In the current study, we found that FGF18 was significantly upregulated in hepatic stellate cells (HSCs), and that paracrine secretion of FGF18 decreased apoptosis of hepatocytes during liver IRI by reducing ROS levels. Mechanistically, FGF18 treatment dramatically reduced the mRNA and protein levels of USP16, leading to reduction of the downregulation deubiquitination level of KEAP1 and the activation of Nrf2. In addition, USP16 interact and deubiquitinated KEAP1 with K48-linked ubiquitination. It is worth noting that FGF18-induced Nrf2 directly bound to the promoter of USP16 and thus formed a negative feedback loop with USP16. In conclusion, our findings reveal a function for FGF18 and suggest that FGF18 is a promising target in hepatic IRI.

## Results

### Hepatic FGF18 is upregulated in HSCs after hepatic IRI

Firstly, the involvement of FGF18 in hepatic IRI were analyzed. The public RNA-sequencing showed that FGF18 expression was significantly increased in the livers of mice subjected to IRI (Fig. 1A). Next, we analyzed expression of FGF18 in liver samples of patients who had undergone partial liver resection. Both the mRNA and protein levels of FGF18 were significantly increased after post-resection compared with the baseline levels (Fig. 1B and Fig. S1A). Consistently, the mRNA and protein levels of FGF18 were also markedly elevated in liver lobes of mice subjected to hepatic IRI, and the upregulation was mainly distributed on the ischemic liver lobes. (Fig. 1C and Fig. S1B). Interestingly,

the postoperative FGF18 levels in in patients were positively correlated with ALT levels at POD1 (Fig. 1D: $R^2 = 0.6073$, $p < 0.0001$). Enzyme-linked immunosorbent assays demonstrated that the serum levels of FGF18 were elevated in these mice and patients (Fig. 1E and Fig. S1C). Considering that FGF18 is a member of the paracrine FGF family, and may be secreted by other organs and act on the liver[23], we analyzed FGF18 expression levels in different organs by western blotting. Upon hepatic IRI, FGF18 expression was most significantly altered in the liver among all the organs analyzed, indicating that the liver is the main site of FGF18 expression (Fig. 1F). To further clarify the main cell types that secrete FGF18 in the liver, we extracted four main primary cells from mouse liver and subjected to RT-PCR analysis. The results showed that HSCs were the major cells secreting FGF18, followed by macrophages. And the expression of FGF18 was lowly expressed in endothelial cells. However, FGF18 was markedly reduced in hepatocytes after IRI (Fig. 1G). Furthermore, we analyzed the colocalization of FGF18 with desmin, LYVE-1, F4/80, and albumin (ALB), which are markers of HSCs, liver sinusoid endothelial cells (LSECs), kupffer cells (KCs), and hepatocytes respectively (Fig. 1H). The same results were obtained using livers of patients and Fgf18-tdTomato mice subjected to I/R surgery (Fig. S1D, E). In addition, we explored the expression levels of FGF18 and its major receptor FGFR3 in both L02 and LX-2 cells subjected to hypoxia/reoxygenation (H/R). In accordance with the in vivo results, FGF18 was upregulated in LX-2 cells after H/R stimulation, while FGFR3 was highly expressed in hepatocytes and its expression was unchanged in LX-2 cells (Fig. S1F). Furthermore, transwell experiment showed that hepatocytes exhibited higher levels of apoptosis when co-cultured with FGF18-knockdown HSCs (Fig. S1G), suggesting that HSC-secreted FGF18 acts on hepatocytes upon liver IRI. Next, we investigated the expression of other FGF receptors in hepatocytes subjected to H/R, we found that FGFR1 and FGFR2 levels were almost unchanged, while the expression of FGFR4 was decreased (Fig. S1H). Furthermore, upon treatment with the supernatant of H/R-challenged LX-2 cells, FGFR3-depleted hepatocytes exhibited higher levels of apoptosis than hepatocytes treated with the scrambled control H/R challenge (Fig. S1I). Taken together, these results indicate that activated HSCs secrete FGF18, which affects hepatocytes via FGFR3 upon hepatic IRI.

### FGF18 overexpression alleviates liver damage during hepatic IRI

To clarify the role of FGF18 in regulation of hepatic IRI, we constructed mice that overexpressed FGF18 in the liver by injecting AAV2/9-FGF18 via tail vein. Overexpression of FGF18 in the liver was confirmed by western blotting and immunofluorescence (Fig. 2A and Fig. S2A). The elevated serum levels of ALT and aspartate aminotransferase (AST) at 6 h after I/R surgery were significantly attenuated in FGF18-overexpressing mice compared with wild-type (WT) mice (Fig. 2B). In addition, the necrotic area was considerably smaller in liver sections of FGF18-overexpressing mice, as assessed by hematoxylin and eosin (H&E) staining (Fig. 2C). Apoptosis plays an important role in hepatic IRI, and suppression of apoptosis is a promising strategy to treat IRI. Upon hepatic IRI, the number of apoptotic cells was significantly lower in the livers of FGF18-overexpressing mice than in those of WT mice, as demonstrated by TUNEL staining and c-CAS-3 immumohistochemistry staining (Fig. 2D, E). Moreover, the levels of the antiapoptotic protein Bcl-2 and the proapoptotic genes Bax were significantly higher and lower, respectively, in FGF18-overexpressing mice than in WT mice at 6 h after hepatic I/R surgery (Fig. 2G). Also, inflammatory response is another essential step in the pathogenesis of hepatic IRI. FGF18 overexpression mice exhibited reduced inflammatory response, as verified by immunofluorescence staining of myeloperoxidase (MPO) and analysis of the mRNA levels of IL-6, TNFα, and IL-1β (Fig. 2F, I). Consistent with the in vivo results, FGF18 treatment markedly reduced apoptosis and the inflammatory response in hepatocytes subjected to H/R (Fig. 2H, J). Taken together, these findings suggest that FGF18 protects against IRI-induced apoptosis and inflammation in liver tissue and hepatocytes.

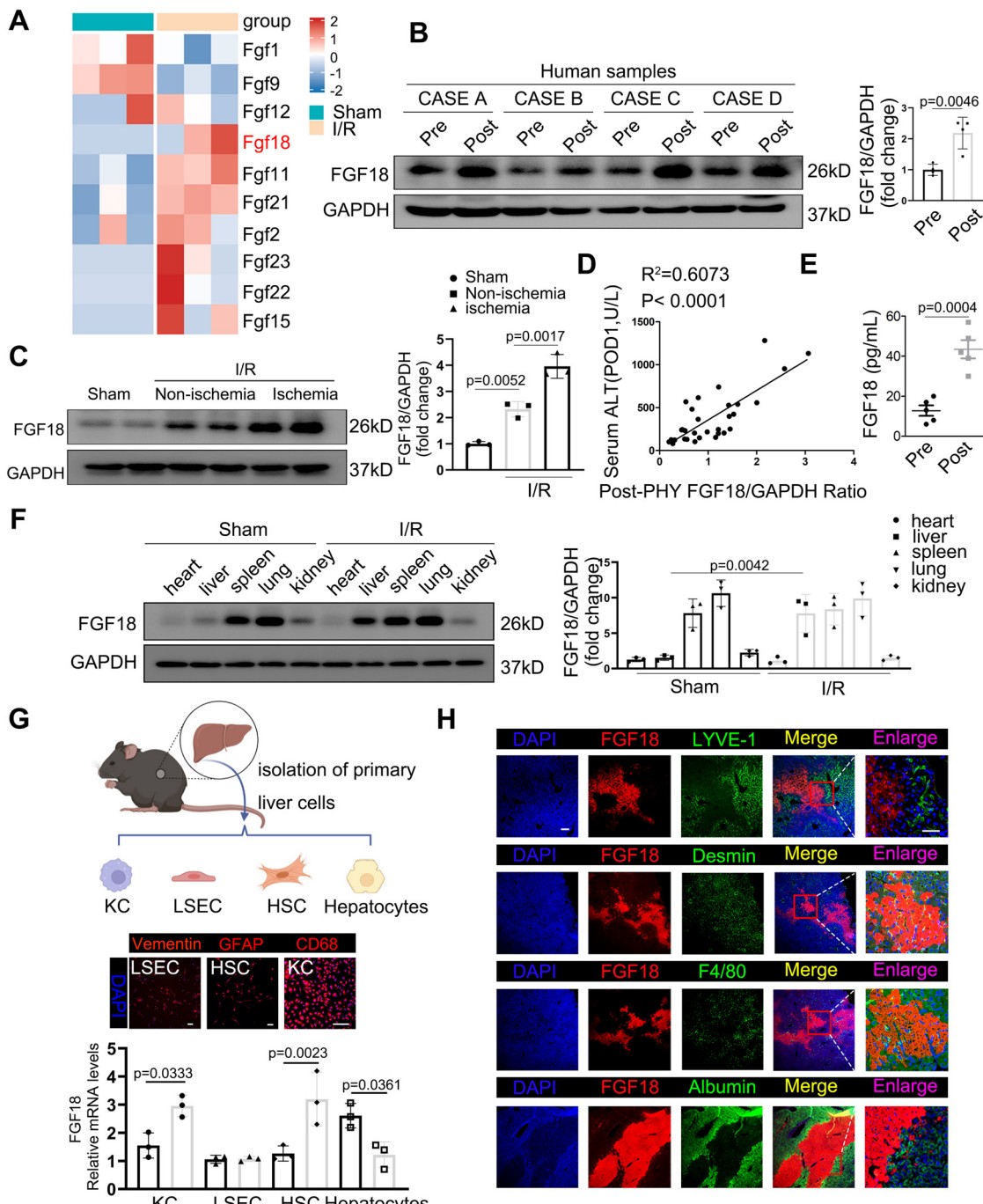

**Fig. 1 | Hepatic FGF18 is upregulated in HSCs after hepatic I/R injury. A** RNA sequence was carried out between I/R and Sham. The results of FGFs changes are presented in the form of heat maps (*n* = 3). **B** The protein expression of FGF18 in human subjected to hepatic resection (*n* = 4). **C** The protein expression of FGF18 in livers of mice subjected to I/R (1 h/6 h) surgery or not (*n* = 3). Non-ischemia represented non-ischemia liver lobes and ischemia represented ischemia liver lobes. **D** The ratio of FGF18/GAPDH after hepatectomy positively correlated positive with ALT at POD1 (*n* = 32). **E** The serum level of FGF18 in human subjected to hepatic resection (*n* = 5). **F** The protein expression of FGF18 in different tissues under hepatic I/R (1 h/6 h) or not (*n* = 3). **G** KCs, LSECs, HSCs, and hepatocytes were extracted from mouse liver and subjected to H/R (4 h/6 h) challenge. All the primary liver cells were verified for purity by immunofluorescence. Scale bar = 200 μm. Then cells were carried out for RT-PCR assay (*n* = 3). (Scheme is Created with BioRender.com) **H** Immunofluorescence for FGF18 (red) and different cell markers (green). Scale bar = 50 μm (*n* = 5). I/R ischemia reperfusion, HSCs hepatic stellate cells, KC Kupffer cell, LSEC liver sinusoids endothelial cell; The statistical significance of differences were assessed by two-tailed student unpaired *t*-test for (**B**, **C**, **E**). Other assays were assessed by one-way ANOVA, followed by Tukey's multiple comparison test. Data are presented as means ± SEM with individual values. All numbers (*n*) are biologically independent experiments. Source data are provided as a Source Data file.

## HSCs-specific FGF18 deficiency aggravates liver apoptosis and the inflammatory response during hepatic IRI

To further evaluate the functional role of FGF18 in HSCs in response to hepatic IRI, GFAP-Cre mice were bred with *Fgf18*^f/f mice to generate HSC-specific conditional FGF18-KO (*Fgf18*^△HSC) mice and the successful knockout of FGF18 was confirmed by isolated HSCs from *Fgf18*^f/f and *Fgf18*^△HSC mice subjected to I/R surgery (Fig. 3A and Fig. S2B). Organs of *Fgf18*^△HSC mice had a normal morphology compared with those of

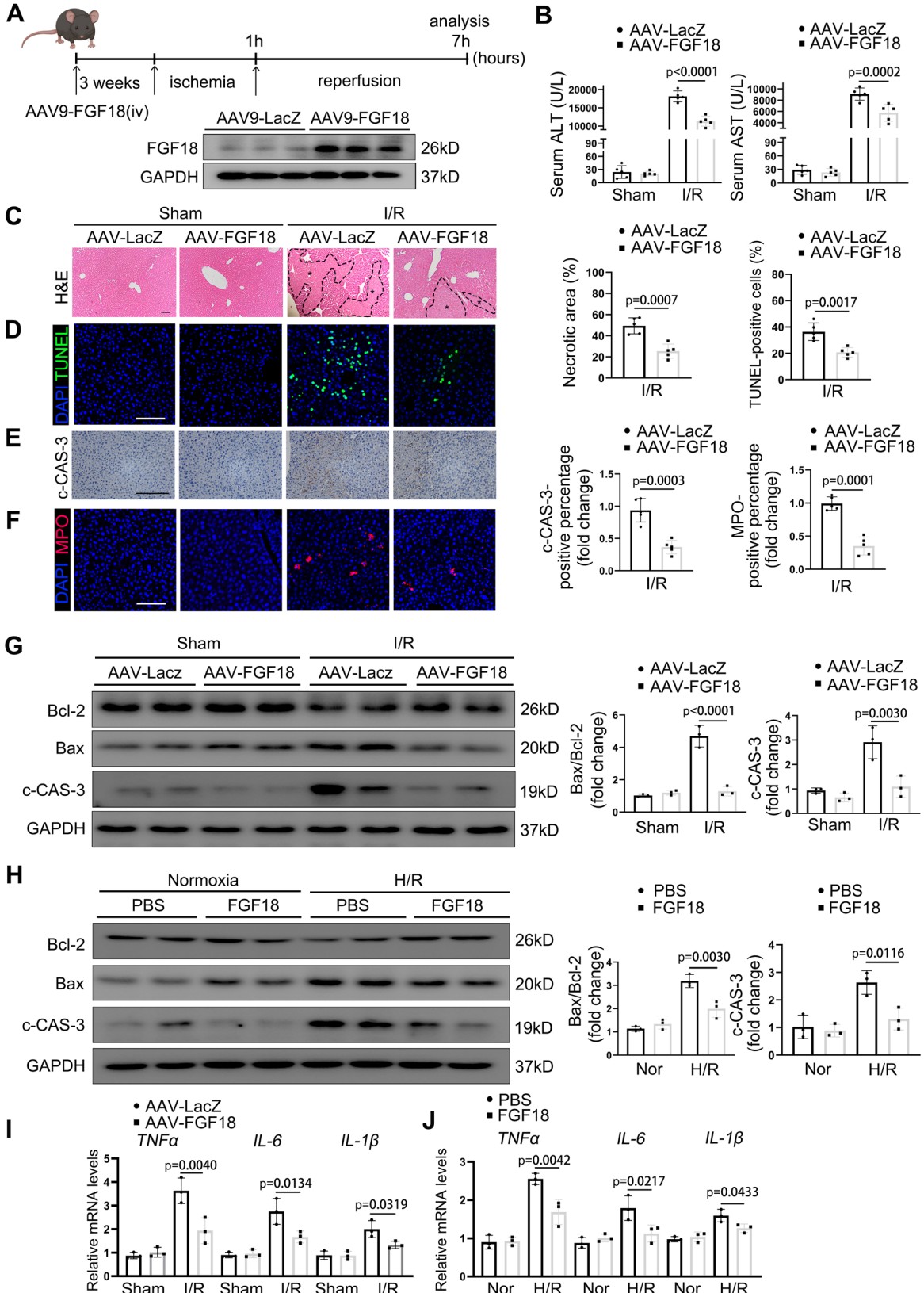

control mice (Fig. S2C). We found that *Fgf18*$^{\triangle HSC}$ mice exhibited significantly impaired liver function upon I/R injury compared with *Fgf18*$^{f/f}$ mice, as indicated by serum ALT and AST levels (Fig. 3B). In addition, *Fgf18*$^{\triangle HSC}$ mice displayed more serious necrotic area than control mice (Fig. 3C), and IRI-induced hepatic apoptosis was aggravated in *Fgf18*$^{\triangle HSC}$ mice, as indicated by TUNEL and c-CAS-3

immunohistochemical staining (Fig. 3D, E). Furthermore, western blotting assays showed that the levels of the antiapoptotic protein Bcl-2 and the proapoptotic protein Bax were lower and higher, respectively, in *Fgf18*$^{\triangle HSC}$ mice than in *Fgf18*$^{f/f}$ mice upon hepatic IRI (Fig. 3G). *Fgf18*$^{\triangle HSC}$ mice also exhibited aggravated inflammatory cell infiltration based on MPO staining (Fig. 3F). Consistent with the increased

**Fig. 2 | FGF18 overexpression alleviates liver damage during hepatic IRI.** Mice were injected with AAV2/9-FGF18 via tail tein. After stably expressed for two weeks, mice were subjected to I/R (1 h/6 h) surgery. **A** Schematic diagram of the work and western blotting showed the transfection efficiency of AAV-FGF18 ($n = 5$). (Scheme is Created with BioRender.com) **B** Serum ALT and AST level ($n = 5$). **C** H&E staining of liver sections. Scale bar=100 μm ($n = 5$). **D–F** TUNEL staining, c-CAS-3 immunohistochemistry, and MPO staining of liver sections ($n = 5$). Scale bar = 100 μm. **G** Relative protein expression by western blotting ($n = 3$). **H** Protein levels of apoptosis related signaling pathway in H/R (4 h/6 h) treated L02 cells in the presence of FGF18 (200 ng/mL) subjected to challenge or not ($n = 3$). **I, J** Relative mRNA expression of genes by RT-PCR ($n = 3$). H/R, hypoxia/reoxygenation; The statistical significance of differences were assessed by two-tailed two-tailed student unpaired *t*-test for **C–F**. Other assays were assessed by one-way ANOVA, followed by Tukey's multiple comparison test. Data are presented as means ± SEM with individual values. All numbers (n) are biologically independent experiments. Source data are provided as a Source Data file.

apoptosis observed in vivo, hepatocytes treated with the supernatant of H/R-challenged LX-2 cells transfected with Ad-sh*FGF18* exhibited higher levels of apoptosis than hepatocytes treated with the supernatant of H/R-challenged LX-2 cells transfected with Ad-Lacz (Fig. S2D). Heatmap analysis further confirmed that FGF18 deficiency significantly exacerbated levels of apoptosis and inflammation compared to those in the WT group (Fig. 3H). Collectively, these in vivo and in vitro results demonstrate that FGF18 deficiency in HSCs promotes apoptosis and inflammation during hepatic IRI.

## FGF18-mediated protective effects in hepatic IRI is USP16 dependent

To determine the mechanisms by which FGF18 regulates apoptosis and inflammation in hepatocytes during hepatic IRI, we focused on an interesting deubiquitination enzyme, USP16. The level of USP16, a deubiquitinase widely implicated to function in cell proliferation, division, and differentiation, was dramatically increased upon liver IRI (Fig. S3). In parallel, the protein and mRNA levels of USP16 were markedly upregulated in clinical samples of patients who had undergone partial liver resection (Fig. 4A, B). Moreover, the postoperative USP16 levels was also positively correlated with ALT levels at POD1 (Fig. 4C: $R^2 = 0.7114$, $p < 0.0001$). Interestingly, FGF18 overexpression significantly decreased the expression of USP16 in mice (Fig. S4A). Consistently, hepatocytes treated with the supernatant of H/R-challenged LX-2 cells transfected with Ad-sh*FGF18* exhibited a higher level of USP16 (Fig. 4D).

Next, we further validated the regulation of USP16 by FGF18 in vivo. $Usp16^{\triangle Hep}$ mice were constructed by injecting AAV-TBG-Cre into $Usp16^{f/f}$ mice via the tail vein. Knockdown of USP16 in the liver was confirmed by western blotting and immunofluorescence staining (Fig. 4E). $Usp16^{\triangle Hep}$ mice were injected with AAV-LacZ or AAV-FGF18 and subjected to IRI-inducing surgery. The increases in apoptosis and the inflammatory response upon I/R surgery were dramatically decreased in $Usp16^{\triangle Hep}$ mice. However, overexpression of FGF18 did not further alleviate IRI in $Usp16^{\triangle Hep}$ mice, as indicated by the similar serum ALT/AST levels, necrotic areas, and levels of tissue apoptosis and inflammation in $Usp16^{\triangle Hep}$ mice injected with AAV-LacZ and $Usp16^{\triangle Hep}$ mice injected with AAV-FGF18 (Fig. 4F–J). Collectively, these results demonstrate that the protective effects of FGF18 against hepatic IRI are dependent on USP16.

## FGF18-mediated protective effects in hepatic IRI are Nrf2 dependent

Next, we performed RNA-sequencing analysis comparing the livers of $Fgf18^{f/f}$ mice and $Fgf18^{\triangle HSC}$ mice subjected to IRI. KEGG and BP analysis showed that FGF18 deficiency induced the profound enrichment of genes associated with xenobiotic and glutathione metabolism (Fig. S5A–D), indicating the activation of Nrf2 signaling. Consistently, IRI-induced expression of Nrf2 and its downstream target HO-1 was much lower in $Fgf18^{\triangle HSC}$ mice than in $Fgf18^{f/f}$ mice (Fig. S5E). In addition, H/R slightly induced Nrf2 expression, whereas FGF18 treatment further increased the total and nuclear levels of Nrf2 in hepatocytes according to western blotting (Fig. S5F). Immunofluorescence analysis yielded the same results (Fig. S5G). DCFH-DA staining showed that FGF18 significantly alleviated ROS levels in H/R treated hepatocytes

(Fig. S5I). Furthermore, transfection of hepatocytes with si-*Nrf2* abolished the protective effects of FGF18 and increased expression of proapoptotic proteins (Fig. S5J).

To verify that amelioration of hepatic IRI by FGF18 is dependent on Nrf2, we generated global *Nrf2 KO* mice and confirmed that Nrf2 was not expressed in the liver using western blotting analyses (Fig. S6A, B). Then, *Nrf2 KO* mice were injected with AAV2/9-FGF18 via the tail vein and subjected to hepatic I/R surgery. *Nrf2 KO* mice with and without FGF18 overexpression showed similar serum ALT/AST levels, necrotic areas, and levels of tissue apoptosis and ROS (Fig. S6C–F, H–I). Moreover, they exhibited similar levels of inflammatory cell infiltration and mRNA levels of the inflammatory factors *TNFα*, *IL6*, and *IL1β* (Fig. S6G, J). Taken together, these results suggest that the Nrf2 signaling pathway is required for the protective effects of FGF18 against hepatic IRI.

## USP16 directly interacts with KEAP1 and prevents the activation of Nrf2 signaling

Considering that FGF18 regulates both Nrf2 and USP16, we speculated that USP16 may mediate the protective effects of FGF18 against hepatic IRI by affecting Nrf2 signaling. As expected, $Usp16^{\triangle Hep}$ mice showed increased protein levels of Nrf2 and HO-1 (Fig. S5K). Also, it is widely known that KEAP1 plays a critical role in the regulation of Nrf2 protein stability. Our experiments found that FGF18 treatment significantly decreased the protein level of KEAP1 along with its unchanged mRNA level (Fig. 5B). Meanwhile, AAV-FGF18-injected mice also exhibited a decreased protein level of KEAP1 (Fig. S4A). In addition, the decreased KEAP1 level induced by FGF18 treatment was reversed in the presence of MG132, but not of CQ, indicating that FGF18 mediates degradation of KEAP1 upon H/R via the deubiquitinase function of USP16 (Fig. 5A). Next, to determine how USP16 affects KEAP1 signaling, co-immunoprecipitation assays and mass spectrometry were performed to detect the potential substrates of USP16 in Flag-USP16-overexpressing HEK293T cells. As expected, KEAP1 was identified as a potential interacting factor (Fig. 5C and Fig. S7A, B). We then examined the interaction between USP16 and KEAP1. Immunoprecipitation assays showed that USP16 and KEAP1 were intrinsically interacted in HepG2 cells under normal conditions (Fig. S7C). Immunofluorescence staining further verified the colocalization of USP16 and KEAP1 in the cytoplasm (Fig. 5D). Proximity ligation assay also yielded the same results (Fig. 5E). Moreover, the interaction between USP16 and KEAP1 was confirmed in HEK293T cells transfected with Flag-USP16 and HA-KEAP1 (Fig. 5F). GST-pull-down assay showed the same result (Fig. 5G). In addition, we mapped the interaction domains between USP16 and KEAP1 mutants, and found that the double glycine repeat (DGR) domain of KEAP1 was crucial for its interaction with USP16 (Fig. 5I).

We then investigated whether USP16 influences the Nrf2 signaling pathway by affecting expression of KEAP1. As expected, H/R dramatically decreased KEAP1 expression, and knockdown of *USP16* further reduced KEAP1 expression, accompanied by activation of the Nrf2 signaling pathway (Fig. 5H). Furthermore, knockdown of USP16 markedly increased KEAP1 degradation rate (Fig. 5J). Collectively, these results demonstrate that USP16 directly interacts with KEAP1 to prevent the activation of Nrf2 signaling.

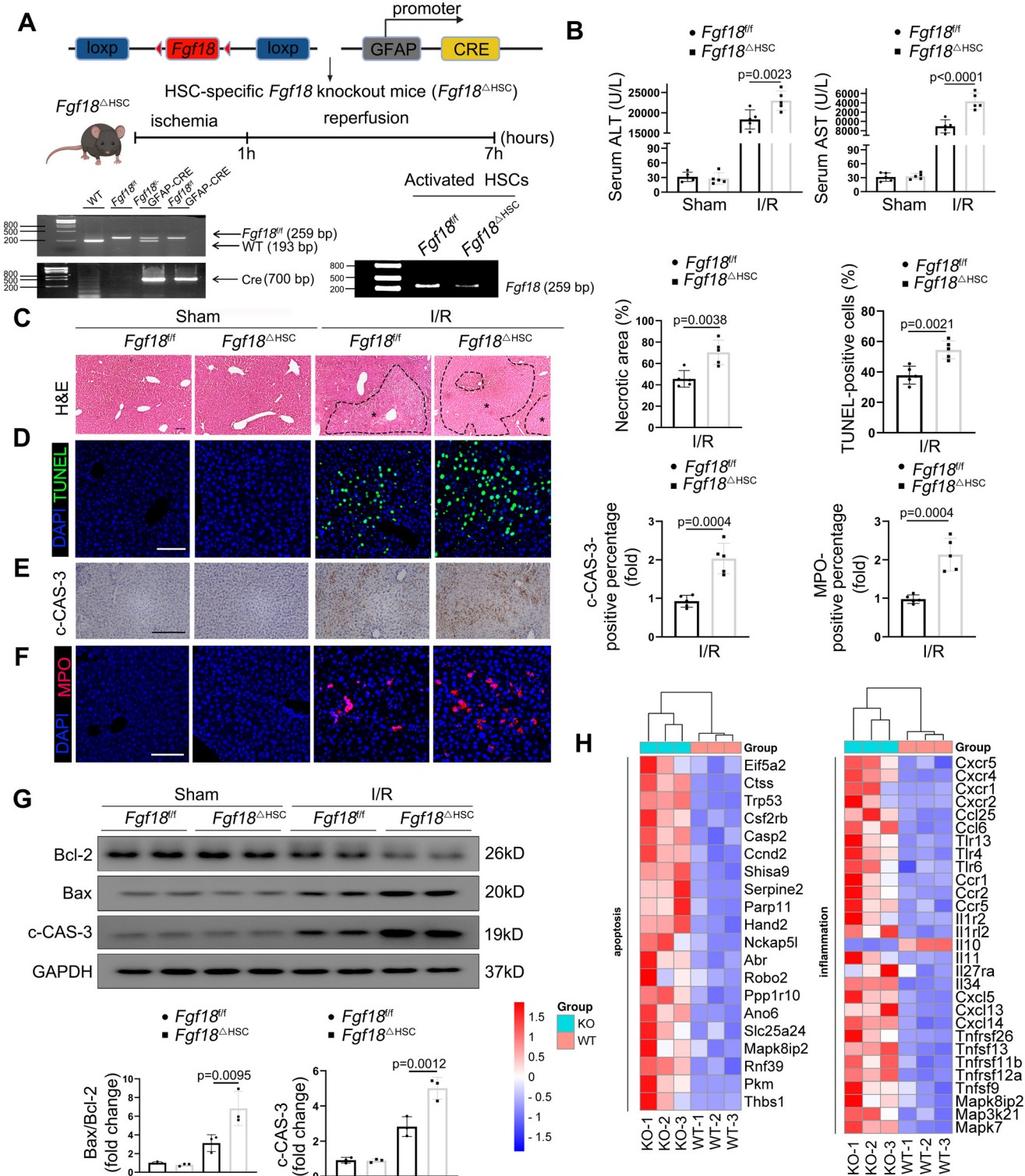

**Fig. 3 | HSC-specific FGF18 deficiency aggravates liver apoptosis and the inflammatory response during hepatic IRI.** $Fgf18^{\triangle HSC}$ mice were constructed by using $Fgf18^{f/f}$ and GFAP-specific Cre mice. **A** Schematic diagram of mouse breeding. And the genotype of $Fgf18^{\triangle HSC}$ mice and the knockdown efficiency of FGF18 were verified by RT-PCR ($n = 3$). (Scheme is Created with BioRender.com). **B** Serum ALT and AST level ($n = 5$). **C** H&E staining of liver sections ($n = 5$). Scale bar = 100 μm. **D–F** TUNEL staining, c-CAS-3 immunohistochemistry, and MPO staining of liver sections ($n = 5$). Scale bar=100 μm. **G** Relative protein expression by western blotting ($n = 3$). **H** RNA-seq analysis of livers from $Fgf18^{f/f}$ mice and $Fgf18^{\triangle HSC}$ mice subjected to I/R (1 h/6 h). Red indicated upregulation while blue indicated downregulation ($n = 3$). The statistical significance of differences were assessed by two-tailed student unpaired $t$-test for (**C**–**F**). Other assays were assessed by one-way ANOVA, followed by Tukey's multiple comparison test. Data are presented as means ± SEM with individual values. All numbers ($n$) are biologically independent experiments. Source data are provided as a Source Data file.

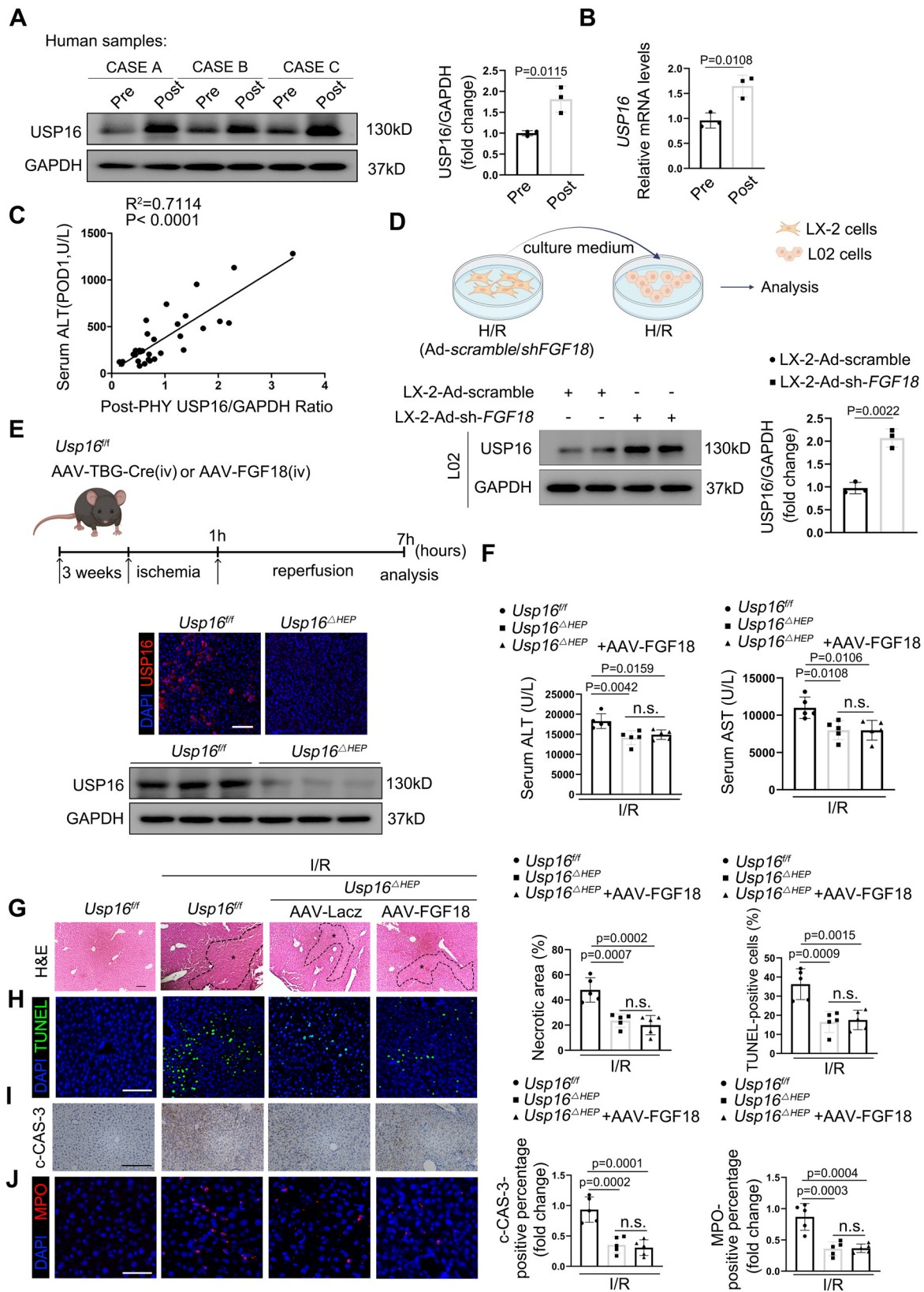

## USP16 inhibits K48-linked ubiquitination and degradation of KEAP1, which are required for alleviation of hepatic IRI by FGF18

To further confirm the role of USP16 in the regulation of KEAP1, we examined the binding between USP16 and KEAP1 upon H/R. The result showed that USP16 indeed interacts with KEAP1, and its interaction was significantly elevated in response to H/R (Fig. 6A). In addition, USP16

knockdown significantly increased ubiquitination of KEAP1. By contrast, USP16 overexpression reduced the ubiquitination of KEAP1, and a catalytically inactive USP16 mutant (C205S) could not deubiquitinate KEAP1 (Fig. 6B, C). The function of USP16 was reportedly dependent on deubiquitination of H2A histones. However, our result showed that the USP16-mediated effects of KEAP1 were independent of H2A histone

**Fig. 4 | FGF18-mediated protective effects in hepatic IRI is USP16 dependent.**
**A** Relative protein expression by western blotting ($n = 3$). **B** Relative mRNA
expression of genes by RT-PCR ($n = 3$). **C** The ratio of USP16/GAPDH after hepa-
tectomy positively correlated with ALT at POD1 ($n = 32$). **D** schematic representa-
tion of the cell-based supernatant transfer assay. Relative protein expression by
western blotting ($n = 3$). (Scheme is Created with BioRender.com) **E** *Usp16*$^{f/f}$ mice
were injected with AAV-TBG-Cre via tail tein. After stably expressed for two weeks,
mice were subjected to I/R (1 h/6 h) surgery. Schematic diagram of the work. And
immunofluorescence and western blotting showed the knockdown efficiency of

USP16 ($n = 5$). (Scheme is Created with BioRender.com) **F** Serum ALT and AST level
($n = 5$). **G** H&E staining of liver sections ($n = 5$). Scale bar = 100 μm. **H–J** TUNEL
staining, c-CAS-3 immunohistochemistry, and MPO staining of liver sections ($n = 5$).
Scale bar = 100 μm. The statistical significance of differences were assessed by two-
tailed student unpaired *t*-test for (**A**, **B**, **D**). Other assays were assessed by one-way
ANOVA, followed by Tukey's multiple comparison test. Data are presented as
means ± SEM with individual values. All numbers ($n$) are biologically independent
experiments. Source data are provided as a Source Data file. (n.s. not significant.).

deubiquitination (Fig. S8). We next examined the subtypes of poly-
ubiquitin chains conjugated to KEAP1, which were removed by USP16
in cotransfected HEK293T cells. USP16 selectively removed K48-linked
polyubiquitin chains from KEAP1 (Fig. S9). Meanwhile, USP16 did not
change the ubiquitination levels of KEAP1 in H/R-challenged HepG2
cells transfected with HA-K48R (Fig. 6D). Considering that USP16 binds
to the DGR region of KEAP1, we further transfected KEAP1-WT or
KEAP1-△DGR in USP16-overexpressing HepG2 cells and challenged
them with H/R. The results showed that USP16 failed to inactivate
Nrf2 signaling pathway, and the expression of proapoptotic protein
was decreased in KEAP1-△DRG HepG2 cells (Fig. 6E). As expected,
FGF18 significantly increased the ubiquitination level of KEAP1 upon H/
R stimulation (Fig. 6F). Overexpression of USP16 completely abrogated
activation of the Nrf2 signaling pathway by FGF18 treatment and its
antiapoptotic effects, while overexpression of the USP16 mutant
(C205S) did not (Fig. 6G). Taken together, these results demonstrate
that USP16 inhibits K48-linked ubiquitination and degradation of
KEAP1, which are required for activation of the Nrf2 signaling by FGF18.

### USP16 is essential for aggravation of hepatic IRI by FGF18 deficiency
Next, to evaluate whether regulation of hepatic IRI by FGF18 is
dependent on USP16 in vivo, we infected *Fgf18*$^{△HSC}$ mice with AAV8-
shUSP16 under the control of TBG promoter to knockdown USP16 in
hepatocytes. The successfully knockdown of USP16 in the liver was
confirmed by western blotting (Fig. 7A). These mice were subjected to
I/R surgery. We found that knockdown of USP16 in *Fgf18*$^{△HSC}$ mice
greatly reduced the necrotic area and improved liver function, as
indicated by H&E staining and measurement of ALT/AST levels
(Fig. 7B, C). Meanwhile, FGF18 deficiency largely increased the apop-
tosis levels in hepatocytes after I/R surgery, and this effect was largely
abolished by the knockdown of USP16 (Fig. 7D, E, H). Moreover,
knockdown of USP16 significantly impaired the inflammatory response
in *Fgf18*$^{△HSC}$ mice, as evidenced by MPO staining (Fig. 7F). As expected,
knockdown of USP16 in *Fgf18*$^{△HSC}$ mice markedly activated
Nrf2 signaling (Fig. 7H) and reduced ROS level (Fig. 7G, I). Taken
together, these results suggest that USP16 is essential for aggravation
of hepatic IRI by FGF18 deficiency.

### FGF18-induced Nrf2 binds to the USP16 promoter and forms a negative feedback loop
The mechanism underlying USP16-mediated KEAP1 deubiquitination
and regulation of Nrf2 signaling was further explored. Interestingly, we
found that both the protein and mRNA levels of USP16 were dramati-
cally increased in ML385 (Nrf2-specific inhibitor)-treated HepG2 cells
challenged with H/R in the presence of FGF18 (Fig. 8A, B). Consistently,
in vivo results also showed that the protein and mRNA levels of USP16
were markedly increased in *Nrf2 KO* mice in response to I/R surgery,
indicating that Nrf2 affects the transcription of USP16 (Fig. 8C, D).
Using the JASPAR database (http://jaspar.genereg.net/), we computa-
tionally found a Nrf2-binding site in the *USP16* promoter region, sug-
gesting that USP16 is a transcriptional target of Nrf2. To identify the
link between Nrf2 and USP16, we performed CHIP assay and found the
significant enrichment of Nrf2 at USP16 promoter under H/R challenge.

And this enrichment was higher in the presence of FGF18 (Fig. 8E).
Moreover, HEK293T cells were cotransfected with a luciferase vector
harboring USP16-WT or USP16 with a mutated promoter region
(Mutant) and Nrf2. After 36 h, a luciferase assay was performed.
Unexpectedly, overexpression of Nrf2 even upregulated the luciferase
reporter expression from USP16-WT, but not from USP16 Mutant
(Fig. 8F). However, in H/R-challenged HepG2 cells, luciferase reporter
expression was markedly induced by H/R while greatly inhibited by
Nrf2 overexpression, indicating that Nrf2 requires the actions of other
transcription factors to repress expression of the *USP16* gene (Fig. 8F).
This topic needs to be researched further.

### Exogenous FGF18 injection alleviates liver damage during hepatic IRI
We further used exogenous FGF18 protein to explore its role in hepatic
IRI (Fig. S11A). As expected, injury was reduced in the livers of FGF18-
treated mice, as reflected by the results of liver function, and western
blotting (Fig. S11B, C). In addition, FGF18-treated mice exhibited less
necrosis area, inflammation and apoptosis (Fig. S11E–G). In summary,
the above results suggested that targeting of FGF18 is a promising
approach to protect the liver against IRI.

## Discussion
Hepatic IRI contributes to the morbidity and mortality associated with
liver resection and transplantation[24]. Therefore, new therapeutic
interventions to alleviate hepatic IRI has been intensely investigated. In
this study, we identified FGF18 as a promising therapeutic target for
hepatic IRI. We found that expression of FGF18 was increased in HSCs
during hepatic IRI in vivo and in vitro. In addition, hepatic IRI was
aggravated by HSC-specific deletion of FGF18 and alleviated by over-
expression of FGF18. Mechanistically, FGF18 decreased expression of
USP16 and thereby upregulated the KEAP1 ubiquitination level, leading
to activation of the Nrf2 signaling pathway. In addition, we found that
Nrf2 regulated USP16 level through transcriptional inhibition. Impor-
tantly, the potential involvement of FGF18 and USP16 in hepatic IRI was
clinically confirmed in clinical patients who had undergone liver
resection. Therefore, FGF18 is a key regulator of hepatic IRI for clinical
applications.

FGF18, a member of the paracrine FGF family, is involved in var-
ious cellular activities. It reduces osteoarthritis by alleviating oxidative
stress and apoptosis in chondrocytes[25]. Furthermore, FGF18 is exten-
sively involved in development of the bones and lungs, and may play a
role in distal lung remodeling rather than having an effect on lung
morphogenesis[26, 27]. Our previous study demonstrated that FGF18
protects against liver fibrosis by inhibiting activation of HSCs mediated
by the Hippo signaling pathway[28]. In the present study, we found that
FGF18 was increased upon hepatic IRI, and that overexpression of
FGF18 significantly ameliorated hepatic IRI. We believe that FGF18
production upon hepatic IRI was transient and moderate, and that it
could not reverse the damage caused by hepatic IRI in our experi-
ments. In addition, our experiments confirmed that specific knock-
down of FGF18 aggravated the damage, which well explains why FGF18
was upregulated upon hepatic IRI, accompanied by a protective role.
Given that the beneficial effects of FGF18, it cannot be directly used as

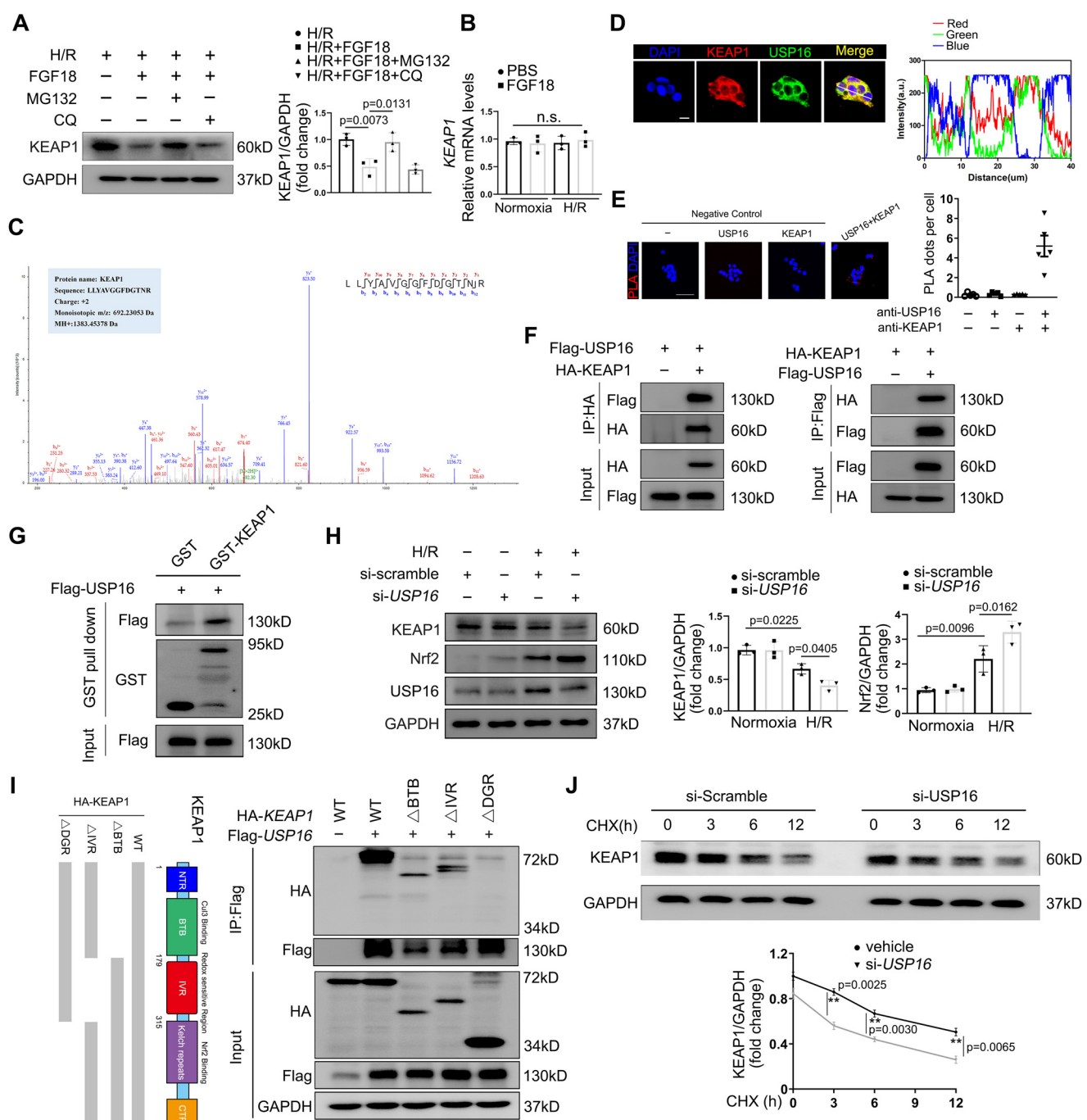

**Fig. 5 | USP16 directly interacts with KEAP1. A** Protein level of KEAP1 was analyzed in HepG2 cells treated with FGF18 or not under H/R (4 h/6 h) challenge. The above cells were simultaneously subjected to MG132 (10 μM) or CQ (50 μM) in the process of H/R (4 h/6 h) (n = 3). **B** Relative mRNA expression of genes by RT-PCR (n = 3). **C** The peptide sequences of the KEAP1 protein was detected by mass spectrometry. **D** Immunofluorescence for KEAP1 (red) and USP16 (green). Scale bar = 20 μm. The white line represented the statistical position (n = 5). **E** PLA assays (n = 5). Scale bar = 50 μm. **F** Representative Co-immunoprecipitation experiment of KEAP1 and USP16 in HEK293T cells (n = 3). **G** Co-immunoprecipitation experiment was carried out to determine the specific binding domain between Flag-USP16 and HA-KEAP1 (n = 3). **H** Relative protein expression by western blotting (n = 3). **I** In vitro GST pull-down assay analysis of the interaction of USP16 and KEAP1 (n = 3). **J** HepG2 cells transfected with si-*USP16* were treated with CHX, and the cells were collected at the indicated times (n = 3). The statistical significance of differences were assessed by one-way ANOVA, followed by Tukey's multiple comparison test. Data are presented as means ± SEM with individual values. All numbers (n) are biologically independent experiments. Source data are provided as a Source Data file. (n.s. not significant.).

an indicator of the extent of hepatic IRI, but we speculated that FGF18 was indeed involved in this process and plays a crucial role. In addition, we also found that FGF18 not only suppressed apoptosis, but also regulated ferroptosis, pyroptosis, and necroptosis (data was not shown). However, the specific mechanism involved in these cell death pathway mediated by FGF18 treatment needed to be further explored.

RNA-sequencing analysis demonstrated that glutathione metabolism was the most significantly enriched pathway upon FGF18 treatment, indicating that the protective role of FGF18 is related to the Nrf2 signaling pathway. Meanwhile, our study showed that FGF18 treatment markedly activated the Nrf2 signaling pathway in vivo and in vitro. To make the H/R model more convincing, we also determined

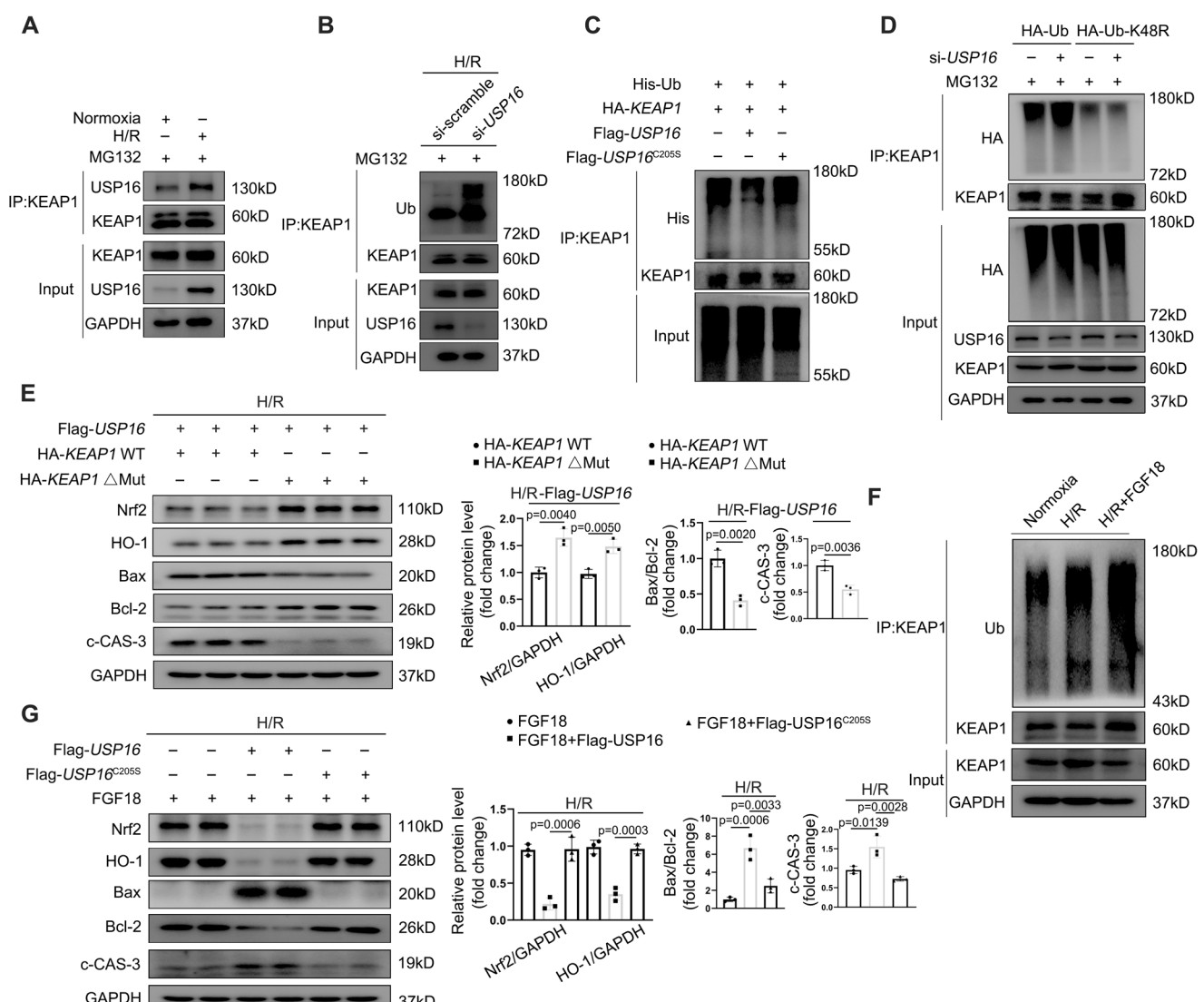

**Fig. 6 | USP16 inhibits K48-linked ubiquitination and degradation of KEAP1, which are required for alleviation of hepatic IRI by FGF18. A** The interaction of KEAP1 and USP16 in HepG2 was detected in H/R (4 h/6 h) challenge or not ($n = 3$). **B** The ubiquitination level of KEAP1 was determined in the indicated group in HepG2 cells ($n = 3$). **C** The ubiquitination level of KEAP1 was determined in the indicated group in HEK293 cells ($n = 3$). **D** HA-Ub or HA-Ub (K48-Mut) was transfected into HepG2 cells in the presence of si-*USP16* or si-*scramble*. The ubiquitination level of KEAP1 was detected in the indicated group ($n = 3$). **E** Flag-USP16 overexpressed HepG2 were transfected with HA-KEAP1 (WT) or HA-KEAP1 (domain Mut). Relative protein expression by western blotting. **F** The ubiquitination level of KEAP1 was determined in the indicated group in HepG2 cells ($n = 3$). **G** HepG2 cells were subjected to H/R (4 h/6 h) challenge in the presence of FGF18. Then cells were simultaneously transfected Flag-USP16, Flag-USP16$^{C205S}$, or not. Relative protein expression by western blotting ($n = 3$). The statistical significance of differences were assessed by two-tailed student unpaired $t$-test for (**E**). Other assays were assessed by one-way ANOVA, followed by Tukey's multiple comparison test. Data are presented as means ± SEM with individual values. All numbers ($n$) are biologically independent experiments. Source data are provided as a Source Data file.

the optimal timing of the H/R models in different hepatocytes (Fig. S10A–C). Nrf2 plays an important role in hepatic IRI[29,30], but its upstream regulatory mechanisms have not been fully elucidated. Intriguingly, our study revealed that the mRNA and protein levels of USP16 were dramatically decreased in the presence of FGF18. USP16, a deubiquitinase, is involved in a variety of cellular activities[31]. Our study further showed that USP16 affected the Nrf2 signaling pathway by directly binding to and deubiquitinating KEAP1 with K48-linked ubiquitination. USP16 has been widely studied as a histone H2A deubiquitinase that modulates gene expression by promoting H2A histone deubiquitination[32]. Surprisingly, FGF18 treatment did not change the ubiquitination level of H2A histones in this study, indicating that FGF18-mediated protection was not due to the histone H2A deubiquitination modification caused by USP16, but by function in a non-classical manner. In addition, USP16 bound to several proteins by mass

spectrometry (supplementary materials), suggesting that the role of USP16 in hepatic IRI was not merely dependent on KEAP1, and thus needs to be studied further. Moreover, We also performed RNA-sequencing between I/R and I/R + AAV-FGF18 in mice, and found that USP2 and USP29 markedly changed after FGF18 overexpression. This may provided another mechanism for FGF18 treatment to improve hepatic IRI. All in all, this is the study to demonstrate that the relationship between FGF and deubiquitinated proteins in the process of hepatic IRI. These findings increase understanding of the role of FGFs in hepatic IRI.

As an activating transcription factor, Nrf2 translocates into the nuclear and promotes the expression of anti-oxidative stress proteins[33]. Interestingly, our work found that Nrf2 elevated luciferase reporter expression from USP16-WT, but inhibited luciferase reporter expression from USP16 under H/R conditions, suggesting that

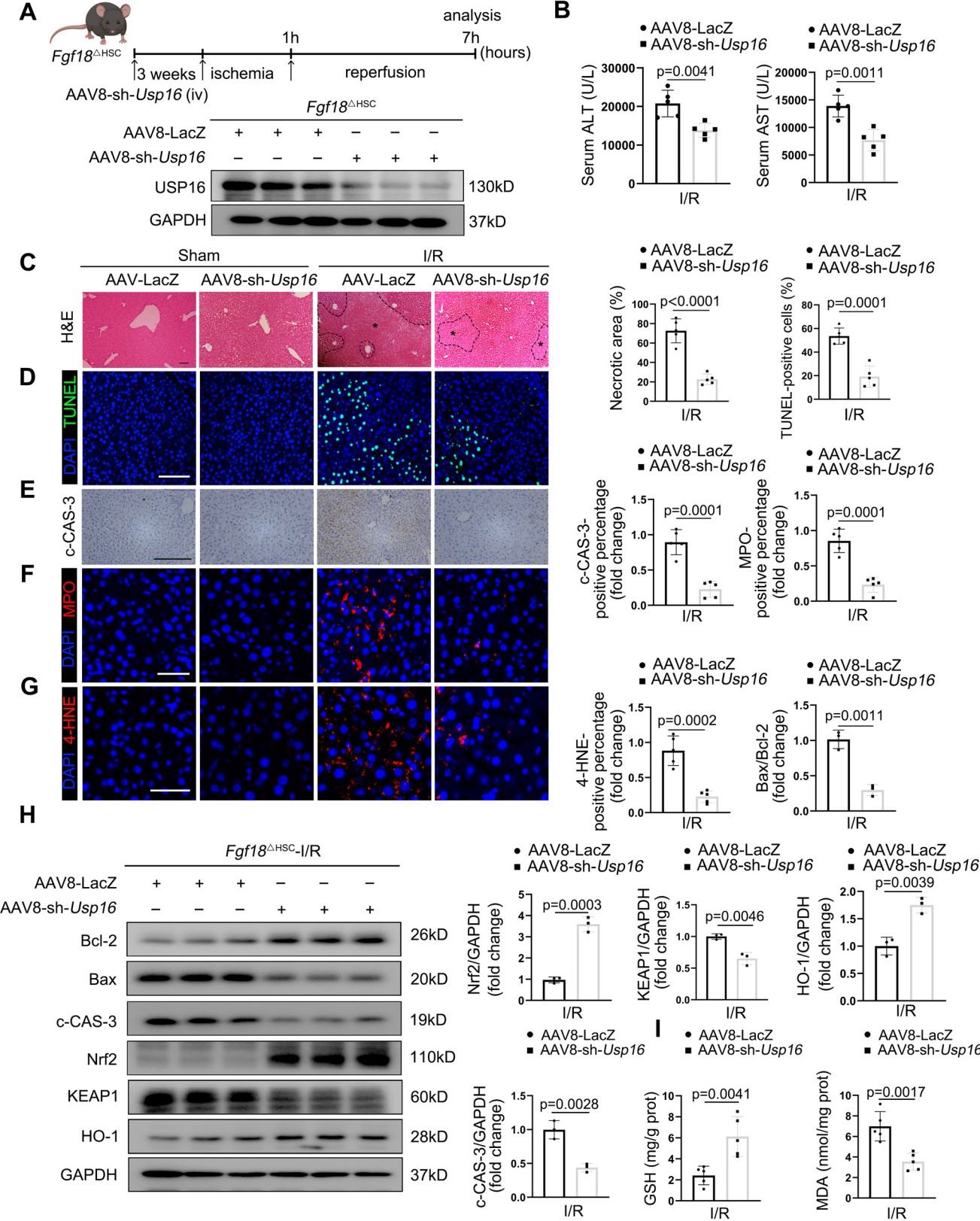

**Fig. 7 | USP16 is essential for aggravation of hepatic IRI by FGF18 deficiency.**
$Fgf18^{\triangle HSC}$ mice were injected with AAV8-TBG-sh-$USP16$ via tail tein. After stably expressed for two weeks, mice were subjected to I/R (1 h/6 h) surgery. **A** Schematic diagram of the work ($n = 5$) (Scheme is Created with BioRender.com). **B** Serum ALT and AST level ($n = 5$). **C** H&E staining of liver sections. Scale bar = 100 μm. **D–G** TUNEL staining, c-CAS-3 immunohistochemistry, MPO, and 4-HNE staining of liver sections ($n = 5$). Scale bar = 100 μm. **H** Relative protein expression by western blotting ($n = 3$). **I** Liver GSH and MDA levels ($n = 5$). The statistical significance of differences were assessed by two-tailed student unpaired $t$-test. Data are presented as means ± SEM with individual values. All numbers ($n$) are biologically independent experiments. Source data are provided as a Source Data file.

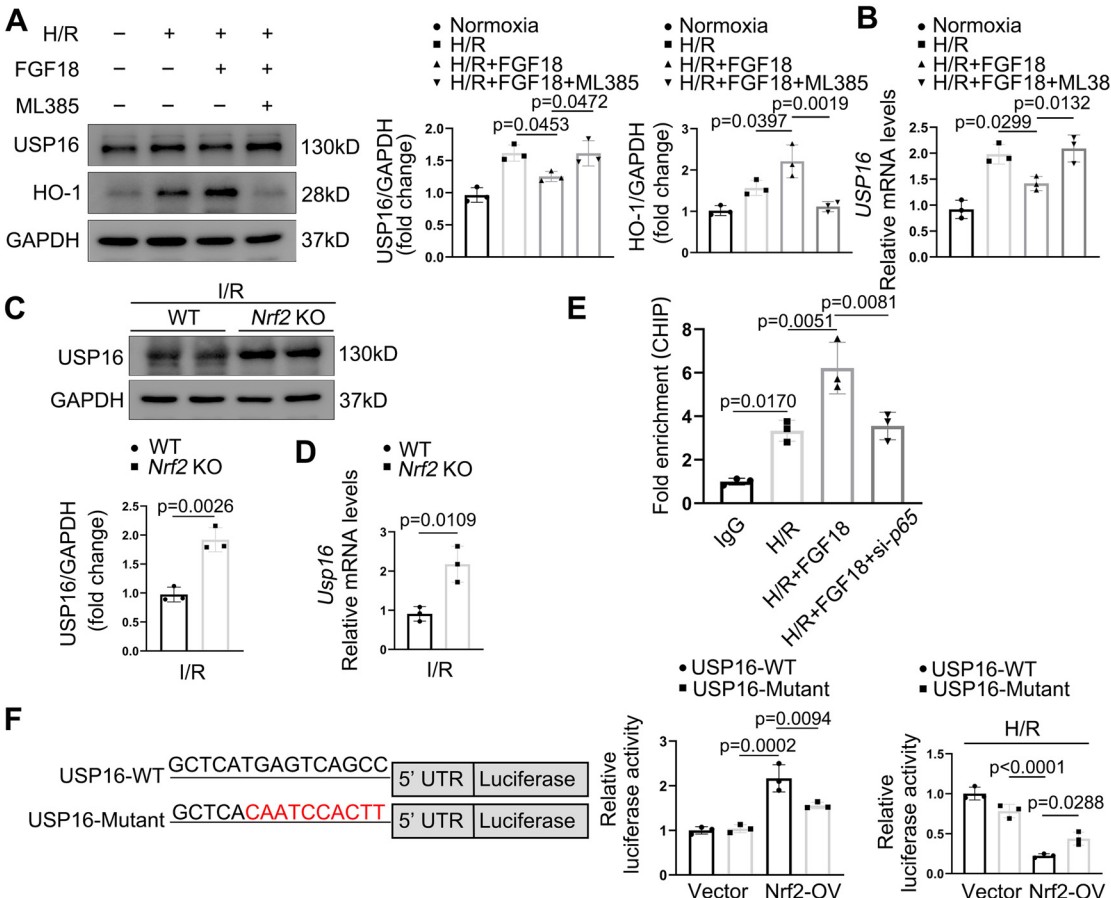

**Fig. 8 | FGF18-induced Nrf2 binds to the USP16 promoter and forms a negative feedback loop. A** Protein levels of USP16 and HO-1 in H/R (4 h/6 h) treated HepG2 cells in the presence or absence of FGF18 or ML385 (10 μM), or not ($n = 3$ per group) ($n = 3$). **B** Relative mRNA expression of genes by RT-PCR ($n = 3$). **C** Relative protein expression by western blotting ($n = 3$). **D** Relative mRNA expression of genes by RT-PCR ($n = 3$). **E** Relative luciferase activity was detected ($n = 3$). **F** Binding of Nrf2 to the USP16 promoter region in H/R treated HepG2 cells was examined by CHIP assay ($n = 3$). The statistical significance of differences were assessed by two-tailed student unpaired $t$-test for (**C**, **D**). Other assays were assessed by one-way ANOVA, followed by Tukey's multiple comparison test. Data are presented as means ± SEM with individual values. All numbers ($n$) are biologically independent experiments. Source data are provided as a Source Data file.

Nrf2 suppresses the mRNA level of *USP16*, and it may need other transcription factors to achieve the functions. Thereby, further researches are still need to be explored. Taken together, our experiments uncovered a noncanonical mode-of-function of Nrf2 during hepatic IRI, consistent with the previous finding that Nrf2 inhibits the inflammatory response by blocking transcription of proinflammatory cytokines[34], in which IL-6 expressed was inhibited by Nrf2 via interfering with the p65-mediated transcriptional activation of the gene. It worth to be note that previous study also demonstrated that p65 can induce the expression of *USP16*[35].

Considering our previous findings that FGF could reduce the expression of p65[36,37], we further explored how FGF affected USP16. As expected, we found that p65 was indeed involved in the regulation of USP16 by FGF18. The enrichment of Nrf2 at USP16 promoter was dramatically reduced by the transfection of *si-p65* (Fig. 8E), suggesting that FGF18 could suppress the USP16 expression via p65. Moreover, previous studies revealed that the expression of USP16 also could be influenced both by LncRNA and MiR-146a[38,39]. Thus, whether FGF18 could affect USP16 through LncRNA and miR-146a needed to be further explored.

Collectively, our data demonstrate that FGF18 protects against hepatic IRI mainly by reducing expression of USP16 and subsequently increasing ubiquitination of KEAP1, and thereby activating Nrf2 signaling. Moreover, we revealed that attenuation of USP16-restrained Nrf2 signaling by FGF18 occurs through a feedback loop involving USP16 and Nrf2. Targeting the FGF18/USP16/KEAP1 regulatory axis is a strategy to alleviate hepatic IRI and related pathological processes.

## Methods
### Reagents
Human FGF18 was produced by Wenzhou Medical University Gene Engineering Laboratory and was stored as powder form at −80 °C. Mouse FGF18 was purchased from Sino Biological (50177-M08H, China). LY294002 (CAS: 154447-36-6) was purchased from Selleck Chemicals (USA). ML385 (CAS: 846557-71-9) was purchased from Selleck Chemicals (USA). MG132 (CAS: 1211877-36-9) was purchased from Selleck Chemicals (USA). CQ (CAS: 54-05-7) was purchased from Selleck Chemicals (USA). Male C57BL/6 mice were purchased from Shanghai Slac Laboratory Animal Co. Ltd.

### Patients and clinical study
Liver clinical samples were obtained from ten patients with benign liver disease undergoing hepatectomy. Pre-hepatectomy hepatic biopsies were harvested after laparotomy (prior to hepatic portal occlusion) and post-hepatectomy hepatic biopsies were obtained after reperfusion (prior to abdominal closure). Ischemic time was from 15 to 40 min. The collection samples were frozen in liquid nitrogen immediately and then stored at −80 °C. Alanine aminotransferase (ALT) was measured on the first day postoperatively (POD1) to assess the degree

of liver injury. Informed consent forms were signed by all subjects. None of the donor organs were obtained from executed prisoners or other institutionalized persons. All liver samples were obtained under protocols approved by the 2nd Affiliated Hospital of Wenzhou Medical University and the information about the patients was summarized in Supplementary Data 1. (Institutional Review Board approval number 2021-K-106-01).

### Animals

Fgf18-LoxP mice (*Fgf18*^f/f, purchased from Cyagen, Suzhou, China) were bred with GFAP-specific Cre mice (GFAP-Cre, purchased from Shanghai Model Organisms Center, Inc.) to create HSC-specific FGF18-KO mice (*Fgf18*^△HSC). *Nrf2 KO* mice was purchased from the Jackson Laboratory (017009). Hepatocyte-specific knockout (*Usp16*^△HEP) mice were generated by injecting *Usp16*^f/f mice with adeno-associated viral-thyroxine binding globulin promoter-Cre (adeno-associated virus (AAV)-TBG-Cre, $3 \times 10^{11}$ virus/200 μL PBS) by tail vein. *Fgf18-tdTomato* and *Usp16*^f/f mice was a kindly gift from Professor Jingling Shen from Wenzhou University. All mice were on a C57BL/6 background. Male C57BL/6 mice were purchased from Shanghai Slac Laboratory Animal Co. Ltd. All mice were housed in a specific pathogen-free (SPF) environment maintained at constant temperature ($22 \pm 2\,°C$) and humidity (40 - 60%) with a 12/12 h dark-light cycle (lights on at 7:00 a.m.), with free access to food and water. All experimental procedures and methods were approved by the Institutional Animal Care and Use Committee of Wenzhou Medical University.

### Mouse liver I/R injury model

A nonlethal and stable mice model generated by 70% partial hepatic warm I/R was used as previously described[40]. Briefly, mice were anesthetized and subjected to midline laparotomy to expose the liver. Next, the portal vein branches that supply the left and middle lobes in the liver were occluded with an atraumatic microvascular clamp. After ischemia for 1 h, the clamp was removed and liver was subjected to reperfusion injury for the indicated times. The exogenous FGF18 protein (10, 20 μg/kg) was injected intraperitoneally into the mice one hour before ischemic surgery and then injected again at the beginning of reperfusion. The relative controls were injected with PBS. Animals were sacrificed after reperfusion. Liver tissues and serum samples were obtained for the further study.

### Cell culture and hypoxia/reoxygenation (H/R) model

HEK293T cells (CL-0005), L02 cells (HL-7702), and HepG2 cells (CL-0103) were purchased from Procell. HEK293T cells and L02 cells were grown in 1640 medium supplemented with 10% FBS. HepG2 cells were grown in MEM (Procell, PM150410P) supplemented with 10% FBS.

Primary hepatic stellate cells were isolated from livers as the protocol previously reported[41]. Mice were first perfused with calcium-free HEPES for 5 min, followed by an in situ collagenase (type IV; Sigma) technique. Primary hepatocytes were separated by centrifugation at $50 \times g$ for 5 min and the supernatants were centrifugation at $600 \times g$ for 10 min. Primary hepatocytes were resuspend with 10 mL DMEM medium mix with 90% Percoll. The cells were plated at 6-well culture dishes with 1640 medium supplemented with 10% FBS and cultured in a 5% $CO_2$/water-saturated incubator at 37 °C. And the HSCs were resuspended in cold GBSSB contained NycodenZ (1002424; Accurate Chemical) at the concentration of 9.69%. Gently overlay with 1.5 mL of GBSS/B and then centrifuge the suspension at $1380 \times g$ for 17 min. The obtained HSCs were resuspended in 1640 medium supplemented with 10% FBS and cultured in a 5% $CO_2$/water-saturated incubator at 37 °C.

For liver sinusoidal endothelial cells (LSEC) and Kupffer cells, after the removal of hepatocytes, centrifuge cell suspension at $300 \times g$ for 10 min. Aspirate supernatant completely. Resuspend cell pellet in 90 μL of buffer per $10^7$ total cells. Add MicroBeads UltraPure (130-092-

007, 130-110-443, Miltenyi Biotec) of LSEC and macrophages respectively and incubated for 15 min. The cells were cleaned by adding 1 mL buffer and centrifuge it at $300 \times g$ for 10 min. Finally, the corresponding cells were obtained by magnetic separation using the LS column of Miltenyi Biotec. Cells were cultured with 1640 medium supplemented with 10% FBS.

To mimic the I/R model in vitro, cells were subjected to H/R. LX-2 cells were placed under hypoxic conditions (1% oxygen) in glucose-free DMEM (11966025; Gibco) for 4 h and then cultured under normoxic conditions for 6 h. L02 cells and primary hepatocytes were placed under hypoxic conditions (1% oxygen) in glucose-free DMEM for 4 h and then cultured under normoxic conditions for 6 h as previously described[42]. HepG2 cells were placed under hypoxic conditions (1% oxygen) in glucose-free DMEM for 5 h and then cultured under normoxic conditions for 2 h as previously described[43]. All the primary liver cells were placed under hypoxic conditions (1% oxygen) in glucose-free DMEM for 4 h and then cultured under normoxic conditions for 6 h.

### Transwell assay

For co-culture system, $1 \times 10^5$ L02 cells were placed in the upper chamber of six-well (Corning) with 0.4 μm pore size, LX-2 cells were placed in the lower chamber. LX-2 cells were transfected with si-*scramble* or si-*FGF18*, then the 6-well was subjected to H/R.

### Cell-based supernatant transfer assay

LX-2 cells were performed with H/R challenge in the presence of Ad-sh*FGF18* or not. Then the culture medium supernatant from LX-2 cells were collected at the end of both hypoxia and reoxygenation. Then L02 cells were subjected to H/R by using the supernatant (hypoxia and reoxygenation, respectively) from LX-2 cells.

### Liver function assessment

Serum levels of alanine aminotransferase (ALT) and aspartate aminotransferase (AST) were measured by commercially available kits (Nanjing Jiancheng Bioengineering Institute, China) according to the manufacturer's protocols to assess the extent of liver damage in animals.

### Dihydroethidium (DHE) and DCFH-DA staining

For DHE (Beyotime, S0063) staining, fresh liver sections were stained with DHE (10 μM) dissolved in PBS for 30 min. Images were observed and captured with a Nikon Eclipse Ni light microscopy (JAPAN). For DCFH-DA (Beyotime, S0033S) staining, L02 cells and HepG2 cells were incubated with Serum-free DMEM medium containing DCFH-DA (10 μM) for 20 min. Then images were observed and captured with Nikon inverted fluorescence microscopy (JAPAN).

### Malondialdehyde (MDA) and glutathione (GSH) assays

Hepatic MDA and GSH levels were analyzed by diagnostic kit (Nanjing Jiancheng Bioengineering Institute, China) according to the manufacturer's protocols.

### Histology and terminal deoxynucleotidyl transferase dUTP nick end labeling (TUNEL) staining

Paraffin liver sections (5 μm in thickness) were deparaffinized and rehydrated, and treated antigen retrieval (6 min, 96 °C). For histopathology, liver sections were stained with hematoxylin and eosin (H&E) (Solarbio, G1120). Images were observed and captured with a Nikon Eclipse Ni light microscopy (JAPAN). For TUNEL staining, paraffin liver sections were performed using a commercially available in situ apoptosis detection kit (Roche Molecular Biochemicals) according to the manufacturer's protocol. And the images were visualized and captured with a Leica SP8 confocal microscopy.

## Mass spectrometry analysis

HEK293T cells were transfected with FLag-USP16, and cells were performed with immunoprecipitation by Flag antibody or IgG (CST, 3900, USA) with protein A/G beads (Meck, LSKMAGAG10, USA). Co-immunoprecipitation was carried out in HEK293T cells and visualized with silver staining by using the beyotime fast silver stain kit (P0017S). Then mass spectrometry was used to detect all the proteins connected with Flag-USP16. The assay was carried by Shanghai Bio-profile Technology Company Ltd. In brief, the gradient consisted of 5-35% (v/v) acetonitrile in 0.1% (v/v) formic acid at a flow rate of 200 nL/min for 10 min, 35-100% (v/v) acetonitrile in 0.1% (v/v) formic acid at a flow rate of 200 nL/min for 2 min and 100% acetonitrile in 0.1% formic acid at a flow rate of 200 nL/min for 8 min. The eluted peptides were ionized and introduced into a Thermo Fisher LTQ Velos Pro mass spectrometer (Thermo Fisher Scientific, Bremen, Germany) using a Proxeon nanoelectrospray ion source. Survey full-scan MS spectra (from m/z 200-1800) were acquired. A full mass spectrum (m/z 200-1800) was followed by fragmentation of the ten most abundant peaks, using 35% of the normalized collision energy for obtaining MS/MS spectra. All peptide assignments were verified by manual inspection. The information about the proteins connected with Flag-USP16 was summarized in Supplementary Data 2.

## RNA-seq analysis and Kyoto Encyclopedia of Genes and Genomes (KEGG) pathway enrichment analysis

RNA-seq was performed at Guangzhou Epibiotek Co., Ltd., (Guangzhou, CHINA). Total RNA was isolated using Trizol reagent (Invitrogen). VAHTS Stranded mRNA-seq Library Prep Kit for Illumina V2 (Vazyme Biotech, NR612-02) was used for library preparation according to the instructions. Reads were aligned to the human Ensemble genome GRCh38 (mouse Ensemble genome GRCm38) using Hisat2 aligner (v2.1.0) under parameters: "--rna-strandness RF". The reads mapped the genome were calculated using featureCounts (v1.6.3). Differential gene expression analysis was performed using the DESeq2 R-package. The information of public RNA-seq was shown in the Supplementary Data 3.

KEGG pathway and Gene Ontology (GO) terms enrichment analysis of differentially expressed genes (DEGs) was carried out by clusterProfiler R Bioconductor package with a $p$-value < 0.05 as statistically significant cutoffs.

## Real-time RT-PCR

Total RNA from liver tissues or cells were extracted by Trizol reagent (Takara Bro Inc, 9108) according to the manufacturer's instructions. One nanogram total RNA was reverse transcribed to generate cDNA by the Hiscript ® III Reverse Transcriptase kit (Vazyme). The cDNA was then subjected to RT-PCR analysis. GAPDH gene was used as a control to target gene expression. The specific primers sequences are shown in Supplementary Data 4.

## RNA interference

L02 cells were transfected with si-FGFR3 (Santa Cruz, sc-29314) or si-NRF2 (Santa Cruz, sc-37030) or siRNA scrambled (negative) control (si-scramble; Santa Cruz, sc-37007) by using Lipofectamine 2000 for 6 h in Opti-MEM. HepG2 cells were transfected with si-USP16 (Limibio Co., Ltd) by using Lipofectamine 2000 for 6 h in Opti-MEM. LX-2 cells were transfected with si-FGF18 (Santa Cruz, sc-39478) by using Lipofectamine 2000 for 6 h in Opti-MEM. After transfection, L02 cells, HepG2 cells and LX-2 cells were subjected to H/R challenge. All the primers for si-RNA sequence used in this study are shown in the Supplementary Data 5.

## Co-immunoprecipitation and ubiquitination assays

For co-immunoprecipitation assays, cells were lysed with ice-cold IP buffer (P10013J, Beyotime) with PMSF. Total lysates (200 μg) were incubated with primary antibodies (1 μg) or IgG overnight at 4 °C with gentle shaking followed by Protein A/G magnetic beads (Thermo

Fisher Scientific) for 4 h at room temperature. Then the co-immunoprecipitated complexes were boiled with 2 × SDS loading buffer at 95 °C for 10 min, followed by western blotting analyses. For ubiquitination analysis, cells were lysed in IP buffer supplemented with 1% SDS and PMSF. Obtained cell lysates were subsequently boiled for 10 min at 95 °C, diluted to 0.1% SDS with IP buffer, and immunoprecipitated as previously described[44]. The primary antibodies were shown in Supplementary Data 6.

## Luciferase assays

The USP16 promoter: luciferase reporter plasmid was constructed in the pGL3 luciferase vector according to the manufacturer's instructions (OBiO Technology, Shanghai, China). The motif in USP16-WT (GCTCATGAGTCAGCC) is mutated to USP16 Mutant (GCTCA-CAATCCACTT). HEK293T cells or HepG2 cells were transfected with 3 μg of pGL3-USP16-luciferase vector and then transfected with the Nrf2-vector. Transcriptional activity was measured using a Dual-Luci-ferase® Reporter Assay System according to the manufacturer's instructions (OBiO Technology, Shanghai, China).

## Nuclear/cytoplasmic fractionation

Cells were subjected to nuclear and cytosolic fractionation using Nuclear/Cytosolic Fractionation Kit (BioVision, Milpitas, CA, USA), following the protocol recommended by the manufacturer.

## Immunohistochemistry and immunofluorescence staining

For immunohistochemistry, liver sections were permeabilized in PBS with 0.2% Triton X-100 for 15 min at room temperature, then blocked with PBS containing 0.5% bovine serum albumin. After 1 h, the liver sections were incubated with primary antibodies at 4 °C overnight: anti-c-CAS-3 (9661, CST, USA). And then incubated with a biotinylated secondary antibody of anti-rabbit IgG at room temperature for 3 h. Next, the liver sections were treated with Diaminobenzidine (DAB) Histochemistry Kit. Finally, the nuclei were stained with hematoxylin. Images were captured with a Nikon Eclipse Ni light microscopy (JAPAN).

For immunofluorescence, in vivo studies, paraffin liver sections were used. For L02 cells and HepG2 cells, cells cultured on glass coverslips were fixed in 4% paraformaldehyde for 30 min. Paraffin liver sections or cells permeabilized in PBS with 0.2% Triton X-100 for 15 min at room temperature. Blocked as above, and then incubated with primary antibody overnight at 4 °C overnight. And then it was incubated with a secondary antibody of Alexa Fluor 647-conjugated anti-mouse IgG (1:200) (Abcam, ab150119) or Alexa Fluor 488-conjugated anti-rabbit IgG secondary antibody (1:200) (Abcam, ab150077). Finally, the nuclei were stained with DAPI. Images were captured with a Leica TCS SP8 Confocal microscope (Leica, Wetzlar, Germany). The primary antibodies were shown in Supplementary Data 6.

## Proximity Ligation Assay (PLA)

The PLA kit was purchased from Merck (DUO92102-1KT). The directly binding between KEAP1 and USP16 was carried out the manufacturer's instructions.

## Plasmid construction and transfection

The full-length homo KEAP1 cDNA was cloned into pcDNA5-HA to express HA-tagged KEAP1 recombinant proteins and full-length homo USP16 cDNA was cloned into pcDNA5-Flag to express Flag-tagged USP16 recombinant proteins. Expression plasmids encoding truncated KEAP1 (1-315aa, 179-624aa, 1-179aa and 315-624aa) were amplified using PCR and cloned into pcDNA5-HA. All the above plasmids were constructed by OBiO Technology (Shanghai, China) Corp., Ltd.. HA-Ub, His-Ub, and His-Ub with different ubiquitin modification types were constructed by limibio Co. Ltd. Transient transfections for plasmids were performed with Lipo 2000 (Thermo Fisher Scientific, USA) and transfected into HEK293T cells. At the same time, MG132 was treated

for 6 h before collection. After transfected for 48 h, HEK293T cells were harvested for co-immunoprecipitation and ubiquitination assays.

## Adeno-associated virus 8/9 (AAV8/9) vectors construction and injection

For the generation of $Usp16^{\triangle HEP}$ mice, $Usp16^{f/f}$ mice were injected with AAV-TBG-Cre ($3 \times 10^{11}$ virus particles/200 μL PBS) by tail vein. After stably expressed for two weeks, $Usp16^{\triangle HEP}$ mice were performed with further study.

For the generation of AAV9-FGF18 mice, C57BL/6 mice were injected with AAV9-FGF18 ($1 \times 10^{11}$ viral particles/200 μL PBS) via the tail vein. After stably expressed for two weeks, AAV9-FGF18 mice were performed with further study.

For the generation of hepatocyte-specific Usp16 knockout $Fgf18^{\triangle HSC}$ mice, $Fgf18^{\triangle HSC}$ mice were injected with AAV8-TBG-shUsp16 ($2 \times 10^{11}$ virus particles/200 μL PBS) to obtain hepatocyte-specific Usp16 knockout $Fgf18^{\triangle HSC}$ mice. After stably expressed for two weeks, the mice were performed with further study.

For the generation of FGF18 overexpression in $Usp16^{\triangle HEP}$ mice, $Usp16^{f/f}$ mice were injected with AAV-TBG-Cre ($3 \times 10^{11}$ virus particles/200 μL PBS) in combination with AAV9-FGF18 ($1 \times 10^{11}$ viral particles/200 μL PBS) through the tail veil. After stably expressed for 2 weeks, the mice were performed with further study.

## Chromatin immunoprecipitation (CHIP)

CHIP assay was performed using the CHIP Assay kit (P2083S, Beyotime) and carried out following the manufacturer's instructions. DNA enrichment was assessed by RT-PCR using 2 × Taq Master Mix (Vazyme). The primers used in CHIP assay are listed in Supplementary Data 4.

## GST-Pull-down assay

Flag-USP16 plasmids were transfected in HEK293 cells, and cells were lysed with IP lysates. Flag-USP16 was purified by the kit (P9801, Beyotime). GST and GST-KEAP1 were transfected in DH5α cells and were purified by GST beads (P2138, Beyotime) at 4 °C. The purified proteins (GST and GST-KEAP1) were incubated with purified Flag-USP16 proteins overnight at 4 °C. Then, the beads were washed 3 times with IP lysates. The complexes were boiled with 2× SDS loading buffer at 95 °C for 10 min, followed by western blotting analyses.

## Western blotting assays

Liver was obtained from hepatic ischemia reperfusion models. The supernatants from liver tissues or cells were extracted by lysis buffer and protein concentration was determined using Pierce BCA Protein Assay Reagent (Thermo Fisher Scientific, 23228). 30 μg protein extracts were loaded and separated by SDS-PAGE and transferred to PVDF membranes (Merck Millipore, IPVH00010). Membranes were blocked with 5% bovine serum albumin in Tris-buffered saline containing 0.1% Tween 20 (TBST) and incubated with specific primary antibodies overnight at 4 °C. Membranes incubated with either goat-anti-mouse HRP (Abcam, ab6789) or goat-anti-rabbit HRP (Abcam, ab6721) for 1 h at room temperature. Proteins were visualized using an Image Quant LAS 4000 (GE Healthcare) system. The expression of specific antigens was quantified using Image Quant 5.2 software (Molecular Dynamics, Inc.). The primary antibodies are FGF18 (Santa cruz, sc-393471, 1:1000), FGFR1 (CST, 9740S, 1:1000), FGFR2 (CST, 23328S, 1:1000), FGFR3 (CST, 4574S, 1:1000), FGFR4 (CST, 8562S, 1:1000), BCL-2 (Huabio, ET1702-53, 1:1000), BAX (CST 2772S, 1:1000), c-Cas-3 (CST, 9661S, 1:1000), Nrf2 (Proteintech, 16396-1-AP, 1:1000), HO-1 (Proteintech, 10701-1-AP 1:1000), KEAP1 (Proteintech, 10503-2-AP, 1:1000), USP16 (Proteintech, 14055-1-AP, 1:1000), HA (Proteintech, 51064-2-AP, 1:1000), Flag (Proteintech, 66008-4-Ig, 1:1000), His (Santa cruz, sc-8036, 1:1000), Ubiquitin (Santa cruz, sc-8017, 1:1000), H2A (Abclonal, A3692, 1:1000), GST (Huabio, ET1611-47, 1:2000). GAPDH (Santa cruz, sc-393471, 1:1000) is used as a loading control.

## ELISA assays

ELISA assays were performed to evaluate the serum FGF18 levels in human (CUSABIO, K05010174) and mice (JINGMEI BIOTECHNOLOGY, JM-12187M1) according to the manufacturer's instructions.

## Statistical analysis

All statistical analyses in this study was performed with GraphPad Prism 5.0 (San Diego, CA, USA). All data are expressed as the mean ± SEM. Statistical analyses used two-tailed student unpaired $t$-test for comparisons between two groups, and one-way analysis of variance (ANOVA) followed by Tukey's multiple comparison test. for comparisons between multiple groups. $P < 0.05$ is considered to be statistically significant.

## Reporting summary

Further information on research design is available in the Nature Portfolio Reporting Summary linked to this article.

## Data availability

The public RNA-seq used in this study are available in the GSE93034 (USP) and SRP117594 (FGF). The mass spectrometry proteomics data have been deposited to the ProteomeXchange Consortium via the PRIDE partner repository with the dataset identifier PXD045021. The RNA-seq data is deposited in GEO with the accession code of GSE242032. All other data supporting the findings of this study are available within the article and its Supplementary Files. Source data are provided with this paper.

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

## Acknowledgements

This work was supported by the National Natural Science Foundation of China (82070507 to W-T. C., 31671252 to J-L.S., 82200699 to X-X.C.,). The Zhejiang Provincial Natural Science Foundation of China (LZ21H020002 to L-T. J., LY20H160014 to B. Z.,). The Science and Technology Project of Medicine and Health of Zhejiang Province of China (2022RC296 to X-X.C.,). Haihe Laboratory of Cell Ecosystem Innovation Fund (22HHXBSS00006 to X-K.L.,).

## Author contributions

G-Z.T., Y-M.C., X-X.C., and W-T.C., designed the experiments; G-Z.T., Y-M. C., X-X. C., J-F.F., K-X.Z., Z-J.H., S-T.L., J-J.Z., J-J.F., Z-H.W., Z-Y.H., B.Z., and L-T.J., performed experiments and analyzed the data; G-Z.T., J-L.S., C.H., and W-T.C., drafted the paper; J-L.S, provided essential experimental materials; X-K.L., and W-T.C., supervised the study, all authors read and approved the final manuscript.

## Competing interests

The authors declare no competing interests.
