## [Peer Review File · Nature Communications]

FGF18 alleviates hepatic ischemia-reperfusion injury via the USP16-mediated KEAP1/Nrf2 signaling pathway in male miceREVIEWER COMMENTS

Reviewer #1 (Remarks to the Author):

This is an elegant series of experiments using deletion and over expression systems as well as protein-protein interactions to identify a novel function for FGF18 in ischemic induced liver injury in mice. The story is reasonably complete and identifies a number of novel findings. There are some potential discrepancies within the data that should be addressed. Furthermore, the authors should take into consideration how FGF suppresses USP16. Is this entirely through the actions of Nrf or are there other signaling mechanisms such as P65 regulation involved? Specific questions are as follows.

1. Does FGF increase in the non-ischemic liver lobes? In other words, is this a non-specific stress response in the liver.
2. The authors focused on apoptosis, but the changes in the large areas of necrosis suggest that FGF18 may suppress other cell death pathways. This should be discussed.
3. I mentioned about some discussion of how FGF suppresses USP16 would be helpful. Because USP16 is proposed to be upstream of Nrf2, the mechanism appears to involve some other effects mediated by FGF18 signaling.
4. The data in figure 1G demonstrating the localization of FGF18 suggests that FGF is regionally upregulated. In addition, the advanced upregulation within focal areas suggests that this occurs also in hepatocytes. The authors should clarify. What do the rows represent in this figure? Where these replicates?
5. It would be helpful if time of sampling were also provided in the figure legend, so that readings do not need to return to the methods section to have this information.

Reviewer #2 (Remarks to the Author):

Liver ischemia-reperfusion injury (IRI) is a common pathological complication in patients undergoing hepatic resection or liver transplantation, resulting in post-operative liver dysfunction. Although many strategies have been reported to ameliorate IRI in animal studies, there is still no effective therapy to prevent and/or treat IRI in human patients. In this study, Tong et al. reported that AAV-based transduction of hepatocytes with FGF18 alleviates hepatic IRI in mice, suggesting that FGF18 is a promising target for treating hepatic IRI. Combining in vivo mouse genetics with in vitro cell line-based biochemical assays, the authors proposed a model containing the following key elements: (1) hepatic IRI induces FGF18 expression (both mRNA and protein) by hepatic stellate cells; (2) Paracrine FGF18 binds to FGFR3 on hepatocytes, leading to reduced USP16 expression (both mRNA and protein); (3) USP16 naturally binds to KEAP1 and ubiquitinates KEAP1 modified with K48-linked poly-ubiquitin chains; (4) USP16 reduction activates NRF2 by reduced KEAP1 polyubiquitination (and degradation); (5) NRF2 represses USP16 transcription as a negative feedback step.

The strengths of the paper include the potential impact of FGF18 as a therapy for ameliorating liver IRI and the in vivo experimental strategies (CRE-FLOX based conditional genetics and AAV-based somatic cell genetics). However, the relevance and interpretation of cell line-based experiments are undermined by unreliable models and insufficient data quality.

Major points are listed as follows:

1. The authors used L02 extensively as an in vitro model for hepatocytes. However, it is now known that L02 is actually an HeLa contaminant (Hepatology 2022, <https://doi.org/10.1002/hep.32730>; <https://www.procell.com.cn/view/743.html>). This critical issue (cell line misidentification and as a result, relevance of experimental data associated with misidentified cell line) needs to be addressed.
2. The only rationale to examine USP16 is that RNA-seq showed USP16 was upregulated upon liver IRI

(Figure S3). However, even within the USP family, many more USPs were upregulated as well (Figure S3). The lack of a clear logic between two major players in the model raises concerns of the validity of the model.

3. The proposed biochemical connection between USP16 and KEAP1 is not fully supported. The authors presented the mass spectra of a peptide derived from KEAP1 in USP16-FLAG IP from 293T cells (Figure 5C). However, quantification of this peptide in USP16-FLAG IP relative to control IP could not be found. The data regarding changes in KEAP1 poly-ubiquitination, half-life, and steady-state level following USP16 knockdown also appears to be modest, raising concerns on the biological significance of these observed changes (Figure 6).

4. The statement in the abstract "FGF18-induced Nrf2 directly bound to the promoter of USP16" is not supported by available data. The authors instead showed a modest increase of USP16 in NRF2 KO cells relative to WT cells and examined the activity of a putative Nrf2 element in driving reporter gene expression. ChIP is needed to support direct binding of Nrf2 to this element.

5. Many figures and their legends lack sufficient details regarding experimental setup, number of replication and exact statistical methods. Several figures appear to be mis-labeled (to name a few, Figures 6A, S1E, S4, S5F, S7).

Minor points:

1. The authors identified FGF18-expressing cells by co-staining with markers of different cell types in the liver (Figure 1G). However, Desmin and F4/80 staining appears to be ubiquitous with limited signal to noise ratio. The authors are advised to isolate different cell types and use RT-qPCR and/or western blotting to corroborate their findings.

2. Co-localization of USP16 with KEAP1 appears to be minimal (Figure 5D). The nature of the PLA negative control was not specified (Figure 5 E). No statistical information was given for these imaging data.

3. There are many typos and inappropriate word use, such as "open RNA-sequencing", "appropriated post-hoc tests". I believe the author meant "public RNA-sequencing" and "appropriate post-hoc tests".

4. The therapeutic impact of FGF18 could be strengthened by showing FGF18 injection as a prophylactic strategy for hepatic IRI in mice.

Response to Reviewers

Reviewer #1 (Remarks to the Author):

This is an elegant series of experiments using deletion and over expression systems as well as protein-protein interactions to identify a novel function for FGF18 in ischemic induced liver injury in mice. The story is reasonably complete and identifies a number of novel findings. There are some potential discrepancies within the data that should be addressed. Furthermore, the authors should take into consideration how FGF suppresses USP16. Is this entirely through the actions of Nrf2 or are there other signaling mechanisms such as p65 regulation involved? Specific questions are as follows.

1. Does FGF increase in the non-ischemic liver lobes? In other words, is this a non-specific stress response in the liver.

Response: Thank you very much for your suggestion. We compared the expression of FGF18 in sham, non-ischemia, and ischemia liver lobes of mice by western blotting. The results showed that only a few of FGF18 were expressed in the non-ischemia liver lobes compared with a large amount of FGF18 expressed in the ischemia liver lobes. We have added some appropriate depiction and images in the revised manuscript (page 7, line 136, Fig. 1C).

2. The authors focused on apoptosis, but the changes in the large areas of necrosis suggest that FGF18 may suppress other cell death pathways. This should be discussed.

Response: Thank you very much for your suggestion. Our results showed that the livers from FGF18 over-expression mice exhibited lower levels of ferroptosis, pyroptosis, and necroptosis compared with relative control. And the specific mechanism involved in these cell death pathway mediated by FGF18 treatment need to be further explored. We have added some appropriate depiction in the revised manuscript (page 19, line 399-403).

3. I mentioned about some discussion of how FGF suppresses USP16 would be helpful. Because USP16 is proposed to the upstream of Nrf2, the mechanism appears to involve some other affects mediated by FGF18 signaling.

Response: According to your suggestion, we further explored how FGF affects USP16. Previous studies have confirmed that p65 could affect the expression level of USP16 through transcriptional level^[1]. In addition, our previous study found that FGF could reduce the expression of p65^[2,3].As expect, in this study we confirmed that p65 was indeed involved in the regulation of USP16 by FGF18.

The enrichment of Nrf2 at USP16 promoter was dramatically reduced by the transfection of si-p65 (Fig. 8E), suggesting that FGF18 could suppress the USP16 expression via p65. Moreover, previous studies revealed that the expression of USP16 also could be influenced both by LncRNA and MiR-146a [4,5]. Thus, whether FGF18 could affect USP16 through LncRNA and miR-146a

needed to be further explored. We have added some appropriate depiction and images in the revised manuscript (page 17, line 352-355; page 21, line 441-448; Fig. 8E).

Reference

[1] Yang S, Wang J, Guo S, Huang D, Lorigados IB, Nie X, et al. Transcriptional activation of USP16 gene expression by NFκB signaling. *Mol Brain*. 2019;12:120.

[2] Li S, Zhu Z, Xue M, Pan X, Tong G, Yi X, et al. The protective effects of fibroblast growth factor 10 against hepatic ischemia-reperfusion injury in mice. *Redox Biol*. 2021;40:101859.

[3] Chen G, An N, Ye W, Huang S, Chen Y, Hu Z, et al. bFGF alleviates diabetes-associated endothelial impairment by downregulating inflammation via S-nitrosylation pathway. *Redox Biol*. 2021;41:101904.

[4] Liu S, Li H, Zhu Y, Ma X, Shao Z, Yang Z, et al. LncRNA MNX1-AS1 sustains inactivation of Hippo pathway through a positive feedback loop with USP16/IGF2BP3 axis in gallbladder cancer. *Cancer Lett*. 2022;547:215862.

[5] Yang Y, Li J, Geng Y. Exosomes derived from chronic lymphocytic leukaemia cells transfer miR-146a to induce the transition of mesenchymal stromal cells into cancer-associated fibroblasts. *J Biochem*. 2020;168:491-498.

4. The data in figure 1G demonstrating the localization of FGF18 suggests that FGF is regionally upregulated. In addition, the advanced upregulation within focal areas suggests that this occurs also in hepatocytes. The authors should clarify. What do the rows represent in this figure? Where these replicates?

Response: We do appreciate the professional comments. We extracted the four major cells from the mice liver and then subjected to RT-PCR analysis. The results showed that the levels of FGF18 in hepatocytes decreased, the levels of FGF18 in Kupffer cells increased slightly. While the levels of FGF18 in hepatic stellate cells significantly increased, and the levels of FGF18 in liver sinusoidal endothelial cell was unchanged during the process of hepatic ischemia-reperfusion. This result was consistent with our previous findings. Moreover, The box in the figure represents the enlarged area, and the white horizontal line represents the scale. We're sorry for making you confused. We have added some appropriate depiction and images in the revised manuscript (page 7, line 146-148, Fig. 1G).

5. It would be helpful if time of sampling were also provided in the figure legend, so that readings do not need to return to the methods section to have this information.

Response: Thanks for your professional advice. We have provided time of sampling in the revised manuscript.

Reviewer #2 (Remarks to the Author):

Liver ischemia-reperfusion injury (IRI) is a common pathological complication in patients undergoing hepatic resection or liver transplantation, resulting in post-operative liver dysfunction. Although many strategies have been reported to ameliorate IRI in animal studies, there is still no effective therapy to prevent and/or treat IRI in human patients. In this study, Tong et al. reported that AAV-based transduction of hepatocytes with FGF18 alleviates hepatic IRI in mice, suggesting that FGF18 is a promising target for treating hepatic IRI. Combining in vivo mouse genetics with in vitro cell line-based biochemical assays, the authors proposed a model containing the following key elements: (1) hepatic IRI induces FGF18 expression (both mRNA and protein) by hepatic stellate cells; (2) Paracrine FGF18 binds to FGFR3 on hepatocytes, leading to reduced USP16 expression (both mRNA and protein); (3) USP16 naturally binds to KEAP1 and deubiquitinates KEAP1 modified with K48-linked poly-ubiquitin chains; (4) USP16 reduction activates NRF2 by reduced KEAP1 polyubiquitination (and degradation); (5) NRF2 represses USP16 transcription as a negative feedback step.

The strengths of the paper include the potential impact of FGF18 as a therapy for ameliorating liver IRI and the in vivo experimental strategies (CRE-FLOX based conditional genetics and AAV-based somatic cell genetics). However, the relevance and interpretation of cell line-based experiments are undermined by unreliable models and insufficient data quality.

Major points are listed as follows:

1. The authors used L02 extensively as an in vitro model for hepatocytes. However, it is now known that L02 is actually an HeLa contaminant (Hepatology 2022, <https://doi.org/10.1002/hep.32730>; <https://www.procell.com.cn/view/743.html>). This critical

issue (cell line misidentification and as a result, relevance of experimental data associated with misidentified cell line) need to be addressed.

Response: Thank you very much for your suggestion. L02 cells were purchased from Procell in 2018, supplied with STR identification report. To elevate the reliability of the experiment, with the primary hepatocytes was introduced to confirmed the conclusion in this study. Similarly, we simultaneously used L02 cells, HepG2 cells, and primary hepatocytes to verify the protective effect of FGF18 (Fig. S5G, Fig. S10D). Nevertheless, L02 cells are still widely applied in recent published works ^[1,2].

Reference

[1] Xu MX, Tan J, Ge CX, Dong W, Zhang LT, Zhu LC, et al. Tripartite motif-containing protein 31 confers protection against nonalcoholic steatohepatitis by deactivating mitogen-activated protein kinase kinase 7. *Hepatology*. 2023;77:124-143.

[2] Chen S, Lu Z, Jia H, Yang B, Liu C, Yang Y, et al. Hepatocyte-specific Mas activation enhances lipophagy and fatty acid oxidation to protect against acetaminophen-induced hepatotoxicity in mice. *J Hepatol*. 2023;78:543-557.

2. The only rationale to examine USP16 is that RNA-seq showed USP16 was upregulated upon liver IRI (Figure S3). However, even within the USP family, many more USPs were upregulated as well (Figure S3). The lack of a clear logic between two major players in the model raises concerns of the validity of the model.

Response: Thank you very much for your suggestion. we thought it was a limitation of our experiment, please understand our situation. And we added some discussion about it in the new

revised manuscript (page 20, line 423-426).

3. The proposed biochemical connection between USP16 and KEAP1 is not fully supported. The authors presented the mass spectra of a peptide derived from KEAP1 in USP16-FLAG IP from 293T cells (Figure 5C). However, quantification of this peptide in USP16-FLAG IP relative to control IP could not be found. The data regarding changes in KEAP1 poly-ubiquitination, half-life, and steady-state level following USP16 knockdown also appears to be modest, raising concerns on the biological significance of these observed changes (Figure 6).

Response: Thanks for your professional advice. We optimized the experiment of USP16 affecting KEAP1 (Fig. 5J, Fig. 6B). In addition, we performed a mass spectrometry test again. The results showed that the expression of KEAP1 in group of USP16-FLAG was significantly higher than that in control IP group. We have put the results in a new EXCEL table and marked KEAP1 and USP16 in yellow highlighting (mass spectrum data). Moreover, we also performed the GST pull down assay. These results indicated that USP16 could directly bind KEAP1 and play a deubiquitination role. We have added some appropriate depiction and images in the revised manuscript (page 14, line 289, Fig. 5G; Fig. 5J, Fig. 6B).

4. The statement in the abstract “FGF18-induced Nrf2 directly bound to the promoter of USP16” is not supported by available data. The authors instead showed a modest increase of USP16 in NRF2 KO cells relative WT cells and examined the activity of a putative Nrf2 element in driving reporter gene expression. ChIP is needed to support direct binding of Nrf2 to this element.

Response: We do appreciate the very professional comments. We have confirmed that Nrf2 could bind to the promoter region of USP16 through luciferase assay, and CHIP assay further found the

significant enrichment of Nrf2 at USP16 promoter under H/R challenge. And this enrichment was higher in the presence of FGF18 treatment. Taken together, the above results indicated that Nrf2 could regulate USP16 via transcription. We have added some appropriate depiction and images in the revised manuscript (page 17, line 352-355, Fig. 8E).

5. Many figures and their legends lack sufficient details regarding experimental setup, number of replication and exact statistic methods. Several figures appear to be mis-labeled (to name a few, Figures 6A, S1E, S4, S5F, S7).

Response: We do appreciate the very professional comments. We are sorry for make you confused. We have revised the above description in the revised manuscript. In addition, we also revised mis-labeled figures. We are sorry again for these mistakes.

Minor points:

1. The authors identified FGF18-expressing cells by co-staining with markers of different cell types in the liver (Figure 1G). However, Desmin and F4/80 staining appears to be ubiquitous with limited signal to noise ratio. The authors are advised to isolate different cell types and use RT-qPCR and/or western blotting to corroborate their findings.

Response: We do appreciate the very professional comments. We have reduced noise ratio to make the Desmin and F4/80 staining more clearly (Figure 1H). Furthermore, according to your suggestion, we extracted the four major cells from the mice liver and performed RT-PCR analysis. The results showed that the levels of FGF18 in hepatocytes decreased, the levels of FGF18 in Kupffer cells increased slightly. While the levels of FGF18 in hepatic stellate cells significantly increased, and the levels of FGF18 in liver sinusoidal endothelial cell was unchanged during the

process of hepatic ischemia-reperfusion. This result was consistent with our previous findings. Overall the results indicated that hepatic stellate cells are the main source of FGF18 secretion during hepatic ischemia-reperfusion. We have added some appropriate depiction and images in the revised manuscript (page 7, line 146-148, Fig. 1G).

2. Co-localization of USP16 with KEAP1 appears to be minimal (Figure 5D). The nature of the PLA negative control was not specified (Figure 5 E). No statistical information was given for these imaging data.

Response: Thanks for your professional advice. We re-tested the combination of USP16 and KEAP1. At the same time, we also added the negative control of PLA assay. We have added some appropriate depiction and images in the revised manuscript (Fig. 5D-E).

3. There are many typos and inappropriate word use, such as “open RNA-sequencing”, “appropriated post-hoc tests”. I believe the author meant “public RNA-sequencing” and “appropriate post-hoc tests”.

Response: We do appreciate the very professional comments. We are sorry for make you confused. We have revised the above description in the revised manuscript.

4. The therapeutic impact of FGF18 could be strengthened by showing FGF18 injection as a prophylactic strategy for hepatic IRI in mice.

Response: Thank you very much for your suggestion. We investigated the effect of FGF18 on hepatic IRI by intraperitoneal injection into the mice. The results showed that FGF18 administration could attenuate hepatic IRI, which provided a solid theoretical basis for FGF18 treatment in the clinical therapeutics of hepatic IRI. We have added some appropriate depiction

and images in the revised manuscript (page 17, line 364-370, Fig. S11).

REVIEWERS' COMMENTS

Reviewer #1 (Remarks to the Author):

The authors have addressed my concerns adequately.

Reviewer #2 (Remarks to the Author):

I am satisfied with the revision experiments and have no further request.